# Single cell transcriptomic profiling identifies tumor-acquired and therapy-resistant cell states in pediatric rhabdomyosarcoma

Sara G. Danielli [1,8], Yun Wei [2,3,4,8], Michael A. Dyer [5], Elizabeth Stewart [5,6], Heather Sheppard [7], Marco Wachtel [1,9] ✉, Beat W. Schäfer [1,9] ✉, Anand G. Patel [5,6,9] ✉ & David M. Langenau [2,3,4,9] ✉

Rhabdomyosarcoma (RMS) is a pediatric tumor that resembles undifferentiated muscle cells; yet the extent to which cell state heterogeneity is shared with human development has not been described. Using single-cell/nucleus RNA sequencing from patient tumors, patient-derived xenografts, primary in vitro cultures, and cell lines, we identify four dominant muscle-lineage cell states: progenitor, proliferative, differentiated, and ground cells. We stratify these RMS cells/nuclei along the continuum of human muscle development and show that they share expression patterns with fetal/embryonal myogenic precursors rather than postnatal satellite cells. Fusion-negative RMS (FN-RMS) have a discrete stem cell hierarchy that recapitulates fetal muscle development and contain therapy-resistant FN-RMS progenitors that share transcriptomic similarity with bipotent skeletal mesenchymal cells. Fusion-positive RMS have tumor-acquired cells states, including a neuronal cell state, that are not found in myogenic development. This work identifies previously underappreciated cell state heterogeneity including unique treatment-resistant and tumor-acquired cell states that differ across RMS subtypes.

Rhabdomyosarcoma (RMS) is a pediatric solid tumor that shares features with arrested skeletal muscle precursors[1,2]. Pediatric RMS has been classified into three major subtypes that have divergent molecular drivers: (i) fusion-positive RMS (FP-RMS) have DNA translocations that juxtapose *PAX3* or *PAX7* with *FOXO1* (PAX3::FOXO1 or PAX7::FOXO1); (ii) fusion-negative RMS (FN-RMS) lack pathognomonic translocation events, but commonly have oncogenic activation of RAS signaling; and (iii) spindle cell/sclerosing rhabdomyosarcoma (SS-RMS, a subclass previously classified as FN-RMS) are driven either by NCOA2::VGLL2 translocations or a p.Leu122Arg mutation in the *MYOD1*

transcription factor[3–13]. Despite aggressive therapies combining radiation, chemotherapy, and surgery, 70% of patients with unresectable or disseminated disease develop recurrent RMS that has a dismal 5-year overall survival rate of under 20%[14–17].

Like many pediatric cancers, RMS tumors have a low mutational burden, and few known genetic alterations reliably predict recurrent disease[8,18–20]. Thus, it is critical to understand the non-genetic heterogeneity within RMS, and the role that specific cell subpopulations play in driving the clinical behavior of RMS. Indeed, multiple groups have applied single-cell transcriptomics to discover distinct RMS cell

[1]Department of Oncology and Children's Research Center, University Children's Hospital of Zurich, Zürich, Switzerland. [2]Molecular Pathology Unit, Massachusetts General Research Institute, Charlestown, MA, USA. [3]Krantz Family Center for Cancer Research, Massachusetts General Hospital, Charlestown, MA, USA. [4]Harvard Stem Cell Institute, Cambridge, MA, USA. [5]Department of Developmental Neurobiology, St. Jude Children's Research Hospital, Memphis, TN, USA. [6]Department of Oncology, St. Jude Children's Research Hospital, Memphis, TN, USA. [7]Department of Pathology, St. Jude Children's Research Hospital, Memphis, TN, USA. [8]These authors contributed equally: Sara G. Danielli, Yun Wei. [9]These authors jointly supervised this work: Marco Wachtel, Beat W Schäfer, Anand G Patel, David M Langenau. ✉e-mail: marco.wachtel@kispi.uzh.ch; beat.schaefer@kispi.uzh.ch; anand.patel2@stjude.org; dlangenau@mgh.harvard.edu

subpopulations[21–24]. These studies consistently identified malignant cells with expression patterns similar to developing skeletal muscle; yet, each study introduced different nomenclature and classification strategies due to limited sample numbers, differences in bioinformatic approaches, and mapping shared developmental cell states across mouse and/or human muscle. As a result, there is a need to clearly define cell states, to assess developmental similarity between RMS and human muscle, and to evaluate the dynamics of cell state transitions during therapy.

Here, we present a consensus evaluation of intratumoral heterogeneity in human RMS by combining datasets encompassing 72 samples from patient tumors, patient-derived xenograft (PDX) models, PDX-derived primary cell cultures (PDCs), and commercial cell lines (CLs)[21,22,24,25]. By uniformly processing and integrating these datasets, we generated a comprehensive and unified annotation of RMS-specific cell subpopulations. In total, we identified four major RMS cell subpopulations—(1) progenitor cells that are largely quiescent and express characteristic mesenchymal and extracellular matrix genes; (2) differentiated cells that are post-mitotic and resemble mature skeletal muscle; (3) proliferative cells that are actively dividing but lack expression of progenitor or differentiated cell programs; and, (4) ground state cells that lack expression of the other three dominant signatures. While we identified shared RMS cell states with embryonal and fetal skeletal muscle development, we also found subtype-specific cell states. Some FP-RMS contain a unique neuronal cell state, indicating that a subset of FP-RMS acquire non-myogenic gene expression programs during tumorigenesis. In addition, progenitor cells in FN-RMS closely resemble bipotent SkM. Mesenchymal cells found in fetal muscle development, which was not observed in FP-RMS. Both FN-RMS and FP-RMS failed to share similarity with postnatal satellite cells. Together, these results challenge the dogma that RMS follow rigid muscle developmental hierarchies and that RMS originate from or resemble satellite-cell derived post-natal muscle. Finally, we show that our cell state signatures can be used to identify treatment-persistent cell populations. Specifically, progenitor and neuronal signatures were significantly enriched in treated samples in FN-RMS and FP-RMS, respectively. In total, this work presents a harmonized model of intratumoral heterogeneity within RMS and provides insights into the intersection of normal development and therapy resistance within cancer.

## Results

### A single-cell/nucleus transcriptomic atlas of RMS

Several groups have investigated the transcriptional heterogeneity of rhabdomyosarcoma (RMS) using single-cell RNA (scRNAseq) and/or single-nucleus RNA sequencing (snRNAseq). These studies identified RMS cell states using different bioinformatic methods, leading to divergent and often confusing nomenclature[21,22,24]. To overcome these limitations, we collected and uniformly processed 72 scRNAseq or snRNAseq datasets from four previously published studies that used the 10X Genomics platform[21,22,24,25] ($n = 107,523$ malignant cells/nuclei). This unified cohort included tumors and experimental models derived from patients seen across four medical centers worldwide, along with established cell line models. This dataset encompasses the largest transcriptomic atlas for any sarcoma analyzed to date and includes patient tumors ($n = 21$), PDXs (patient-derived xenograft, $n = 32$), PDCs (patient derived cell culture, $n = 14$), and conventional CLs (cell lines, $n = 5$) (Fig. 1A). Forty-five datasets were generated from either patient tumor samples or PDXs from the St. Jude Childhood Solid Tumor Network (CSTN)[26], and an additional 6 PDCs were generated from CSTN xenografts (Supplementary Data 1)[22]. Importantly, these samples are representative of intermediate and high-risk RMS, and include primary, recurrent, and metastatic tumors (Supplementary Data 1). All major subtypes of disease were represented including FP-RMS ($n = 27$, previously known as alveolar RMS), FN-RMS ($n = 43$, previously known as embryonal RMS), and two SS-RMS cases with

$MYOD1^{L122R}$-mutations (Supplementary Data 1). After merging datasets and performing dimensionality reduction, malignant cells/nuclei grouped separately based on patient and model systems, consistent with observations in other cancer types[27–29] (Fig. 1B and S1A).

To identify shared transcriptomic signatures across different samples, we corrected for inter-patient variation utilizing anchor-based integration[30]. Following integration, samples were intermixed and as expected, we did not identify outlier cells/nuclei that were attributable to only one patient, dataset, or model system (Fig. S1B). We next applied unsupervised clustering and identified 12 Louvain clusters that could be grouped into distinct subpopulations based on shared transcriptomic profiles (Fig. 1C, D and S1C). These subpopulations include: (1) two clusters with cells/nuclei expressing varying levels of mesoderm transcription factors (e.g., *MEOX2*), cell surface markers (e.g., *CD44*, *EGFR*), and extracellular matrix proteins (e.g., *FN1*) which we call "progenitor" and "transiting-progenitor" ("TR-progenitor") that were distinguished from each other based on overall levels of marker expression; (2) a "proliferative" subpopulation that comprised four clusters that shared GSEA signature similarity with proliferative and DNA replication gene modules; (3) two clusters of "transiting-differentiated" ("TR-differentiated") and "differentiated" muscle cells/nuclei expressing transcription factors from committed muscle cells (e.g., *MYOG*) and muscle contraction proteins (e.g., *TNNI1*, *MYH8*); (4) an "apoptotic" subpopulation expressing genes associated with cell death (e.g., *BNIP3*); and, (5) "ground" cells that do not show any enrichment of these signatures (Fig. 1C–E and S1C; Supplementary Data 2 and 3). Importantly, we compared five matched PDX samples that were generated by the St. Jude Childhood Solid Tumor Network[26] and that were independently expanded and processed in different labs[21,24]. We noted consistency in cell subpopulation distributions for these samples derived from the same patient, indicating that our analysis was not skewed by experimental setting, xenograft passaging, or protocol differences in cell isolation for scRNAseq (Fig. S1D). We next performed single-cell compositional data analysis (scCODA) that uses Bayesian modeling to quantify differences within single-cell RNA-seq clusters across cohorts[31]. scCODA analysis identified statistically credible increases in progenitor and TR-progenitor cell fractions in FN-RMS while FP-RMS had elevated numbers of TR-differentiated cell states proportions (Fig. S1E).

### A tripartite cell state landscape of RMS

In previous reports, each group identified clusters of cells/nuclei with transcriptomic similarity to muscle-lineage cells[21,22,24]. Despite these similarities, each used different sample types, computational methods to define gene expression signatures, and differing nomenclature for each subpopulation. For example, Patel et al. analyzed 18 matched samples from primary patient and orthotopic PDX models and identified three cell populations which they called mesoderm, myoblast, and myocyte cells based on perceived similarity with mouse muscle development[24]. In their study, the myoblast compartment included both proliferative and non-proliferative cells. In contrast, Wei et al. examined 9 PDX and 4 patient tumors to identify four RMS-specific cell subpopulations that they called mesenchymal, proliferative, differentiated, and ground cells[21]. Finally, Danielli et al. studied 14 PDCs and 3 conventional cell lines to group RMS cells into muscle stem-cell-like cells, cycling progenitors, and differentiated cells[22].

We leveraged our unified RMS cell atlas to refine these cell state signatures. We scored single-cell and single-nucleus profiles within the integrated RMS atlas using published signatures from each prior study. Signatures from all three studies identified similar patterns of heterogeneity in the progenitor-like and differentiated-like cells (Fig. 2A). The one exception was the myoblast signature from Patel et al., which was broadly expressed in most RMS cells or nuclei, and thus does not identify a discernable cell state in RMS. For all other cell states defined in these publications, we detected significant overlap between gene

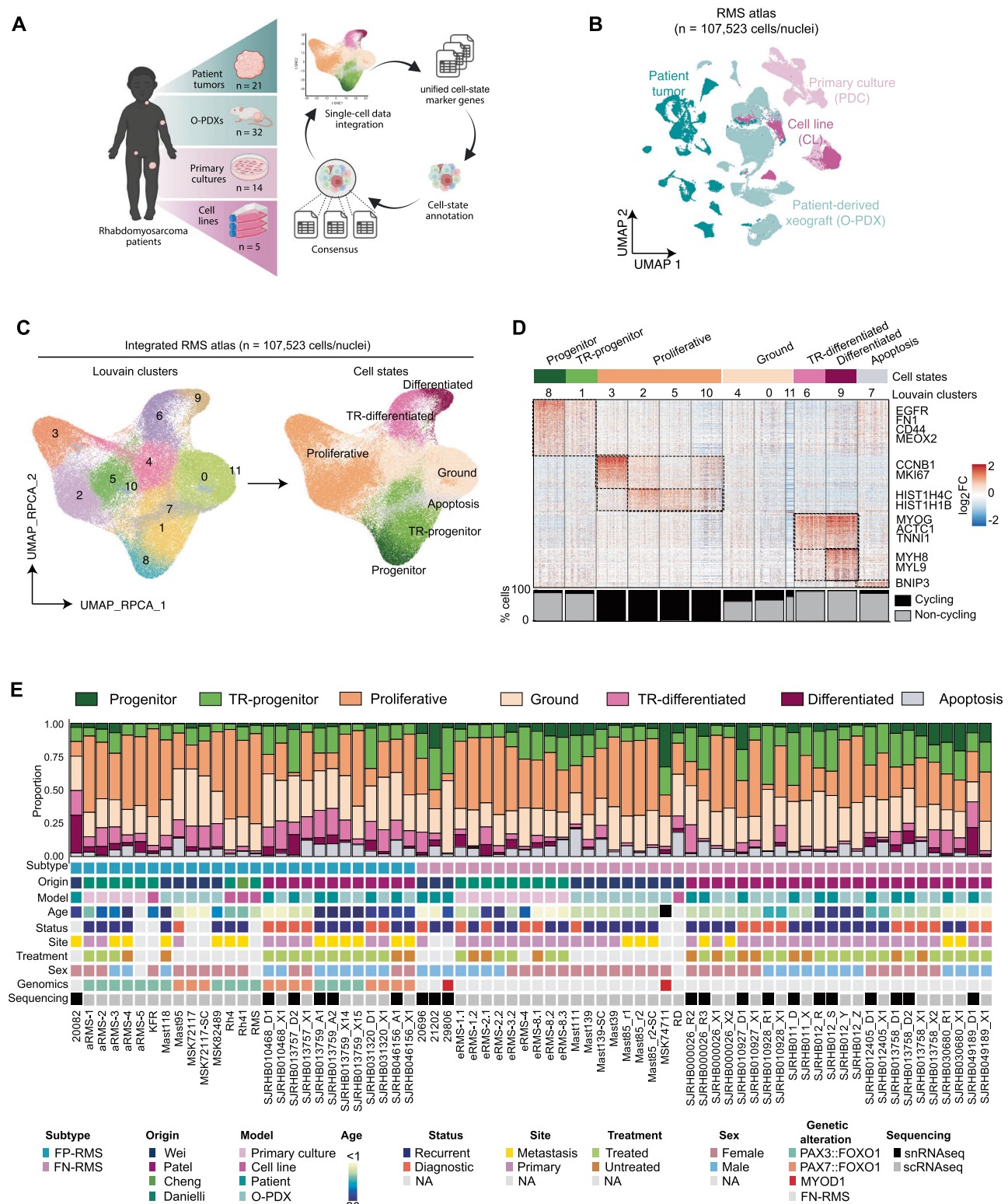

**Fig. 1 | Integrated analysis of single cell and single nuclei sequencing identifies dominant cell states in human RMS.** **A** Schematic of approach and RMS models profiled by single-cell analysis, created with BioRender.com. **B** UMAP plot of tumor cells/nuclei (*n* = 72 datasets) colored by model of origin[21,22,24]. **C** UMAP plot of integrated tumor cells/nuclei using reciprocal PCA (RPCA) projection. Cells/nuclei are colored by Louvain cluster (left) or assigned cell states (right). **D** Heatmap of the genes (x axis) enriched in Louvain clusters (y axis) across the integrated RMS dataset (FC > 0.25; *n* = 400 representative cells/nuclei shown with exception of cluster 11 that contained *n* = 101 cells/nuclei). The percentage of cycling cells within

each cluster is shown in the bar plot below. **E** Summary of the cell state composition of each RMS dataset (*n* = 72) with clinical information included as an oncoplot below. FN-RMS, fusion-negative RMS; FP-RMS, fusion-positive RMS; PDC, PDX-derived primary culture; CL, cell line; PDX, patient-derived xenograft; snRNAseq, single-nuclei RNA sequencing; scRNAseq, single-cell RNA sequencing; NA, not available; TR-progenitor, transiting-progenitor; TR-differentiated, transiting-differentiated. Gene lists used for generating panels D and E are shown in Supplementary Data 2.

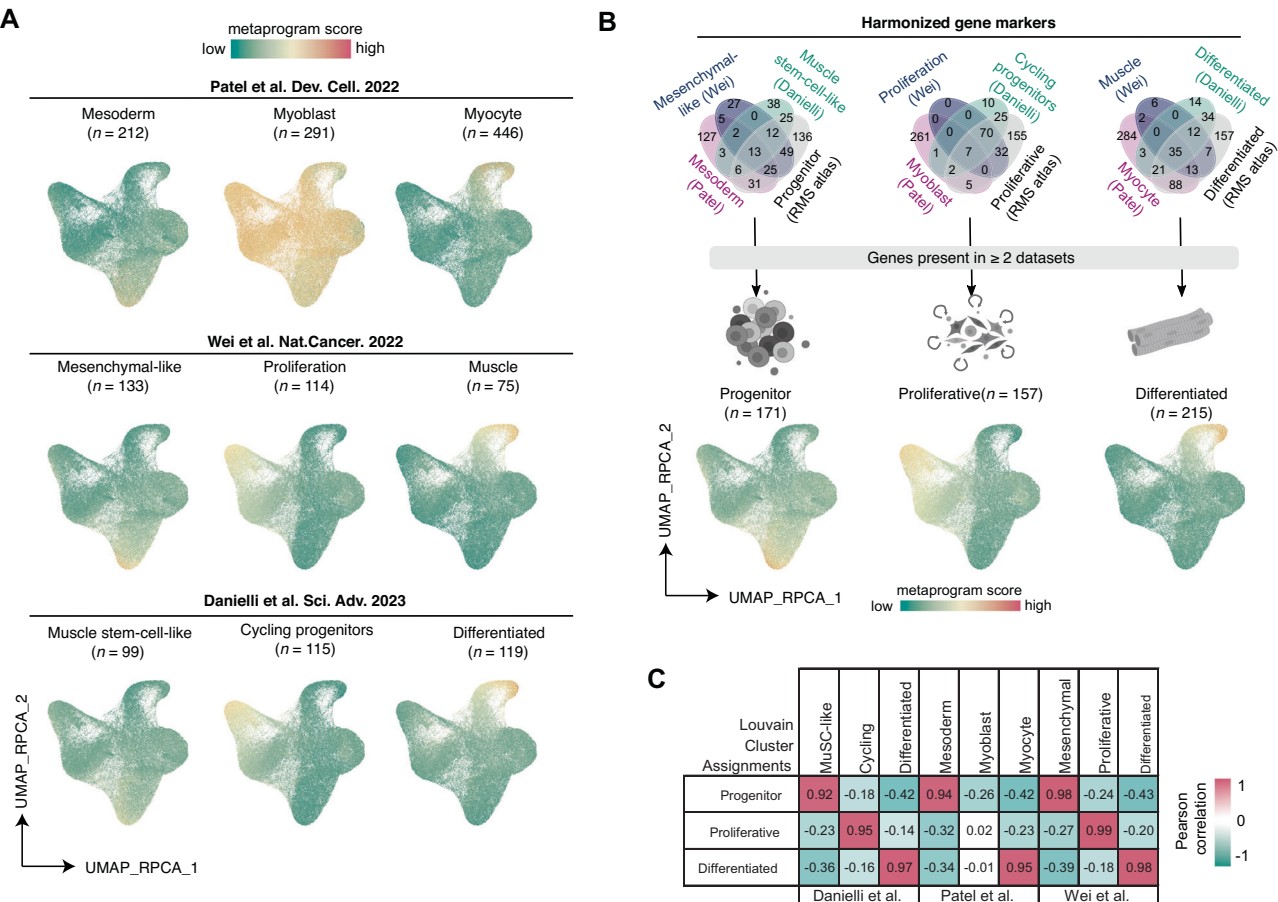

**Fig. 2 | A tripartite cell state landscape of RMS tumors and identification of cell state RMS metaprograms. A** UMAP plots of integrated tumor cells/nuclei (*n* = 107,523 cells) scored for the metaprograms identified in the original publications. Number of genes within each metaprogram noted. **B** Comparison of published cell state metaprograms and those defined by our Louvain clustering approach. Top: Venn diagrams showing overlap of gene markers across the three original publications and our new analysis ("RMS atlas"). Bottom: UMAP plots of integrated tumor cells/nuclei (*n* = 107,523 cells) showing expression of the newly defined, high confidence cell state gene signatures. Number of genes within each metaprogram noted. Icons created with BioRender.com. **C** Pearson correlation coefficients for the metaprograms identified in the three original publications and the new metaprogram signatures defined by our work.

signatures indicating that the previously published studies had independently uncovered similar RMS cell states. These data are also consistent with recently published findings from DeMartino et al. that defined a tripartite cell state landscape in FN- and FP-RMS using a different nomenclature[23].

To construct consensus signatures for the three dominant RMS cell states, we selected genes that were present in at least two datasets to generate signatures for each of the major RMS cell subpopulations. We generated three signatures that we call progenitor (*n* = 171 genes), proliferative (*n* = 157 genes), and differentiated (*n* = 215 genes) signatures (Fig. 2B and Supplementary Data 4). Because these unified gene signature lists were generated from a variety of models (patient tumors, PDXs, PDCs, and CLs), they represent a robust and broadly applicable set of markers for defining each RMS cell state. As expected, these new high-confidence consensus cell state signatures demonstrated significant overlap with those originally reported, with the exception of the Patel, et al. myoblast signature (Fig. 2C).

### The muscle lineage score reveals key distinctions between RMS subtypes

Unsupervised clustering identified previously unknown transitory cells within RMS (Fig. 1C–E), which led us to test whether tumor cell heterogeneity exists within a continuum between progenitor and differentiated RMS cell states. By comparing the progenitor and differentiated signature scores to our categorical cell subpopulations,

we found that both the progenitor and the differentiated signatures showed a gradient of expression across the molecularly defined subpopulations (Fig. S2A). Moreover, most cycling cells/nuclei preferentially mapped to cells with low progenitor and differentiated scores, irrespective of the RMS subtype (Fig. S2B). These results led us to create a "muscle lineage score," defined as the difference between the differentiated and progenitor signature scores, and to apply this scoring metric to every single-cell/nucleus profile within our atlas in relation to their proliferation properties. Indeed, we observed significant inter- and intra-tumoral heterogeneity when stratifying tumor cells using the muscle lineage score. FP-RMS samples had an overall higher muscle-lineage score when compared to FN-RMS, both at the single-cell and pseudo-bulk level (Fig. 3A–C). Also of note, the FP- (*n* = 93 genes) and FN-RMS (*n* = 67 genes) core signatures reported by Wei et al.[21] also separated these two subtypes along a continuous spectrum (Fig. S2C), suggesting underlying gene program differences between these two subtypes of tumors.

To independently validate our results, we next analyzed an additional dataset of 19 single-cell RNA-seq datasets from RMS patients reported by DeMartino et al. that used a plate-based single-cell RNA-seq technique called SORT-seq[23]. Indeed, FN-RMS tumors from DeMartino et al. had higher overall fractions of cells with progenitor signatures, while FP-RMS tumors had elevated numbers of cells with differentiated signatures (Fig. S2D). Moreover, our lineage scoring method validated that FN-RMS tumors consistently had lower overall

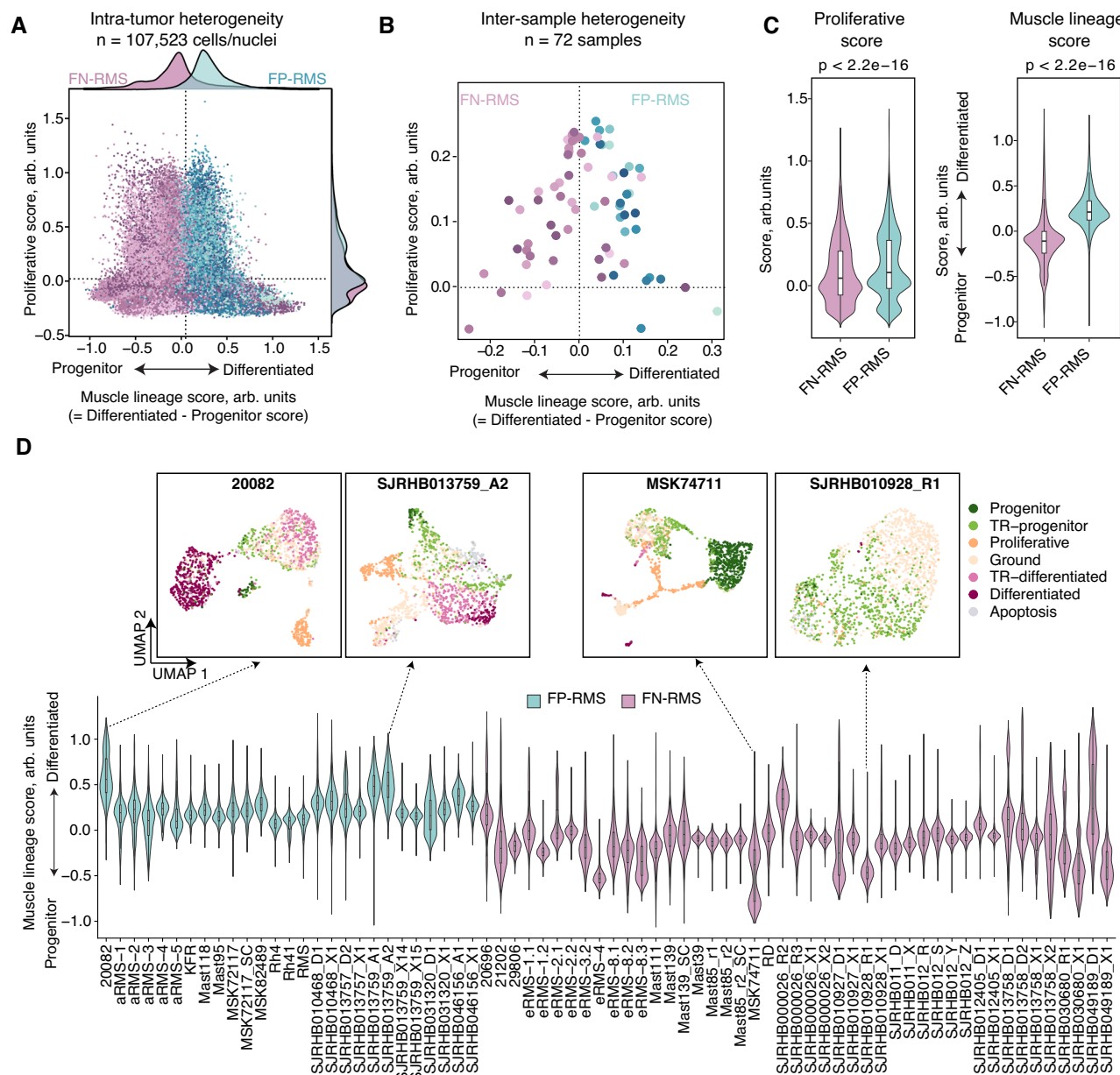

**Fig. 3 | RMS cells lie in a continuum of gene expression defined by three dominant cell states including progenitor, proliferative, and differentiated cell states. A** Graphical analysis showing "muscle lineage score" defined as the difference between the differentiated and progenitor signature scores, and proliferation score of all RMS cells/nuclei. Subtypes are denoted by purple (FN-RMS) and blue (FP-RMS). **B** Average muscle lineage and proliferation scores calculated with pseudo-bulk data for each of the 72 RMS samples. **C** Violin plots showing individual cell expression of proliferative (left) and muscle lineage score (right) across FP-RMS ($n = 40,526$ cells/nuclei) and FN-RMS ($n = 69,997$ cells/nuclei) cells. Boxplots denote Tukey's whiskers (25–75 percentile represented by minima-maxima; statistical median as center). Two-sided student's $t$ test with $p$ values noted in the figure. **D** Violin plots showing cell expression of the muscle-lineage score across each of the 72 RMS samples. Boxplots denote Tukey's whiskers (25–75 percentile represented by minima-maxima; statistical median as center). The UMAP plots of two representative samples for each subtype are shown, with $n = 1500$ cells/nuclei for each individual sample. The number of cells/nuclei analyzed for each sample are reported in Supplementary Data 1.

muscle lineage scores, while FP-RMS tumors had elevated muscle lineage scores (Fig. S2D). In total, our data support a model where RMS cells lie in a continuum of gene expression defined by three dominant cell states including progenitor, proliferative, and differentiated cell states while also containing subtype specific gene programs found within all RMS cells from a given tumor.

Despite overall trends in lower muscle lineage scores in FN-RMS compared with FP-RMS, we did observe considerable inter-tumoral variability of the muscle lineage scores across tumors (Fig. 3D). For example, we identified FP-RMS with exceptionally high muscle lineage scores including 20082 and SJRHB013759_A2. By contrast, FN-RMS

SJRHB010928_R1, a pre-treated FN-RMS, and $MYOD1^{L122R}$-mutant MSK74711 had exceptionally low muscle lineage scores.

### Neuronal cells are a unique feature of FP-RMS

Our initial analyses centered on combining RMS subtypes together to identify conserved cell states shared across pediatric RMS. While this approach enabled us to define key muscle-lineage cell states shared across RMS, it would likely fail to identify subtype-specific subpopulations or differences in gene expression within defined subpopulations in RMS subtypes. Our combined large cohort of RMS samples enabled us to evaluate heterogeneity within FN-RMS, PAX3::FOXO1 FP-RMS, and

PAX7::FOXO1 FP-RMS translocated tumors as distinct entities. As expected, we identified progenitor, proliferative, and differentiated cell subpopulations in each molecular subtype (Fig. 4A and S3A). Yet, we also identified unexpected differences in gene expression between progenitor cell populations from FP-RMS and FN-RMS (Fig. S3B) and two previously unreported gene expression clusters (Supplementary Data 5 and 6). In particular, we found: (1) a group of cells/nuclei in FN-RMS that express interferon response genes such as *ISG15* and *IFIT1-3* ("IFN" cluster; 1.5% of total cells/nuclei); and, (2) a tumor subpopulation in FP-RMS tumors that expresses neuronal marker genes including *DCX*, *L1CAM*, *SYP*, and *CHGA* ('neuronal' cluster; 1.4% and 4.8% of total cells/nuclei from PAX3::FOXO1 and PAX7::FOXO1 FP-RMS, respectively, Fig. 4A, S4A and S4B).

To better characterize the FP-RMS neuronal cell state, we first performed gene set enrichment analysis with highly expressed genes found in this cell cluster and confirmed enrichment of genes associated with neurogenesis pathways including axonogenesis (GO:0007409), central nervous system development (GO:0007417) and central nervous system neuron differentiation (GO:0021953; Supplementary Data 6). Second, we scored each FP-RMS subpopulation for the activity of PAX3::FOXO1 fusion oncogene using a previously defined list of fusion target genes[32]. We found that neuronal cells scored for the highest activity of fusion oncogene activity (Fig. S4C). Last, we performed immunohistochemistry validation on PDX tumor tissue ($n = 5$) and confirmed in situ expression of the neuronal marker synaptophysin (SYP), which correlated with fraction of neuronal cells detected in our single-cell meta-analysis (Fig. 4B, C). Overall, these results confirm that FP-RMS tumors contain a unique subpopulation of tumor cells that expresses markers of neuronal cells. Importantly, cells/nuclei expressing the neuronal gene signature were detected only in a subset of FP-RMS samples ($n = 5$ out of 11 PAX7::FOXO1 FP-RMS and $n = 5$ out of 15 PAX3::FOXO1 FP-RMS, defined as >1% of total cells/nuclei; Fig. S4D and Supplementary Data 7). Despite tumors retaining the tripartite muscle lineage programs across models (Fig. S3C), neuronal subpopulations were detected in larger numbers in patient tumors and PDXs, but rarely or not at all in primary cultures or commercial cell lines [Figs. S4E and S4F; Supplementary Data 7]. The absence of neuronal cells within cell lines may explain why this rare subpopulation has not been deeply investigated before; moreover, the rarity of these cells most likely prevented their identification in prior single-institution single-cell cohorts[23,24].

Lastly, we sought to identify potential candidate cell surface markers in our consensus analysis that could be used to both address future research questions and/or therapeutic targeting. We cross-referenced the gene expression markers identified across each tumor subpopulation for each RMS entity with known cell surface proteins from the Human Protein Atlas. We identified several cell surface markers, including *CD44* for Progenitor, *ERBB3* for Differentiated, and *L1CAM* for Neuronal cells (Supplementary Data 8).

## Mapping shared cell heterogeneity between RMS and human skeletal muscle development

Human skeletal myogenesis proceeds in three defined waves[33]. First, mesodermal progenitors (MPs) from the somite create embryonic myoblasts (MBs) and myocytes (MCs) that drive early skeletal muscle deposition. A second wave of muscle development occurs coincident with the transition of embryonic to fetal development, where both MP and bipotent skeletal muscle mesenchymal progenitor (SkM.Mesen) cells likely drive muscle formation[33,34]. Finally, a third wave of muscle growth is regulated by classically defined muscle stem cells called satellite cells that expand during early postnatal growth to create muscle and then become a reserve stem cell population later in life to aid in repair after injury[35–37]. Here, we took advantage of cell annotations from normal human myogenic development to compare RMS cells/nuclei to these established myogenic cell states[33] (Fig. S5A, B).

We used transfer learning with SingleR, a computational framework that takes a dataset with known labels as an input and then transfers them onto a test dataset based on similarity to the reference[38]. We show that FN-RMS tumor cells shared similarity with a variety of developing human muscle cell types including skeletal mesenchymal cells (SkM.Mesen), myogenic progenitors (MPs), myoblasts (MBs), and myocytes (MCs) (Fig. 4D, E and S5D; Supplementary Data 9). In particular, FN-Progenitor cells preferentially mapped to SkM.Mesen cells and expressed high levels of marker genes of this developmental subpopulation (*OGN*, *THY1*, *POSTN*) (Fig. 4E and S5C). FN-RMS have shared cell states with those found in the second wave of muscle development that starts at week >7 (Fig. S5D). Of note, the number of SkM.Mesen cells peak at 12–14 weeks of development, where they comprise 23.5% of the fetal myogenic cells (Fig. S5B). Similarly, we measured 15.7% of cells/nuclei within FN-RMS mapped to SkM.Mesen cells (Fig. 4E and Supplementary Data 9). This contrasts with FP-RMS that largely lack cell state similarity with SkM.Mesen cells. These results are also in keeping with the identification of differences in gene expression between progenitor cell populations from FP-RMS and FN-RMS (Fig. S3B). PAX3::FOXO1 and PAX7::FOXO1 FP-RMS shared cell state similarity with MBs, MCs and/or myoblasts-myocytes (MB-MCs), with only a minority of cells mapping to MPs, and few to no cells sharing similarity with SkM.Mesen cells (Fig. 4E and S5C; Supplementary Data 9). Finally, RMS do not contain appreciable numbers of cells with shared similarity to postnatal satellite cells and do not map to the third wave of muscle development (Fig. 4E and S5D; Supplementary Data 9). Based on these findings, we propose a refined nomenclature based on shared developmental similarity (or not) with human muscle development: "FN-skeletal muscle mesenchymal-like" (FN-SkM.Mes-like) for FN-RMS Progenitor cell states; and "FP-progenitor" and "FP-neuronal" for cells that represent tumor-acquired cell states in FP-RMS.

## Identification of therapy-resistant cells states in RMS

Treatment recurrence is a major hurdle to achieving durable long-term treatment responses in RMS, and we reasoned that RMS cell states might also define therapy persistent tumor cells. In an effort to identify single-cell signatures that correlate with therapy resistance or tumor recurrence, we compared snRNA-seq datasets from patient FN-RMS samples obtained before therapy ($n = 4$) or amidst therapy ($n = 7$) (Fig. S6A). We observed an enrichment for the Progenitor score and a commensurate reduction in the Differentiated score within treated samples. This observation was particularly pronounced in a pair of samples obtained from the same patient before therapy (SJRHB00026_R2) and amidst therapy (SJRHB00026_R3) (Fig. 5A). Interestingly, the difference was still detectable, though less exaggerated, in orthotopic PDXs generated from those patient samples (SJRHB00026_X1 and _X2; Fig. 5B). Likewise, PDCs generated from another patient obtained before and after therapy (eRMS-8.1, -8.2, and -8.3) showed a similar pattern with a persistent increase in FN-SkM.Mes-like progenitor expression scores (Fig. S6B).

Our observation of cell state shifts within single-cell data from FN-RMS patient samples led us to evaluate a cohort of matched pairs of tumor samples from a single-institution clinical trial, which were obtained from patients who had received a diagnostic biopsy followed by a mid-treatment delayed resection. While a prior study had reported an enrichment for MEOX2 immunopositivity during therapy within these samples[24], we performed RNA-sequencing from fixed clinical samples and applied RMS signatures for the three dominant cell states (Fig. 5C, $n = 9$, [$n = 7$ FN-RMS, $n = 2$ FP-RMS]). In the FN-RMS samples, we detected a significant increase in progenitor scores within the treated patient samples compared to samples from the primary diagnostic biopsy ($p = 0.026$; Wilcoxon signed rank test) and a decrease in proliferative scores ($p = 0.023$; Wilcoxon signed rank test; Fig. 5D). Collectively, these findings demonstrate a treatment-induced selection for the progenitor state in FN-RMS.

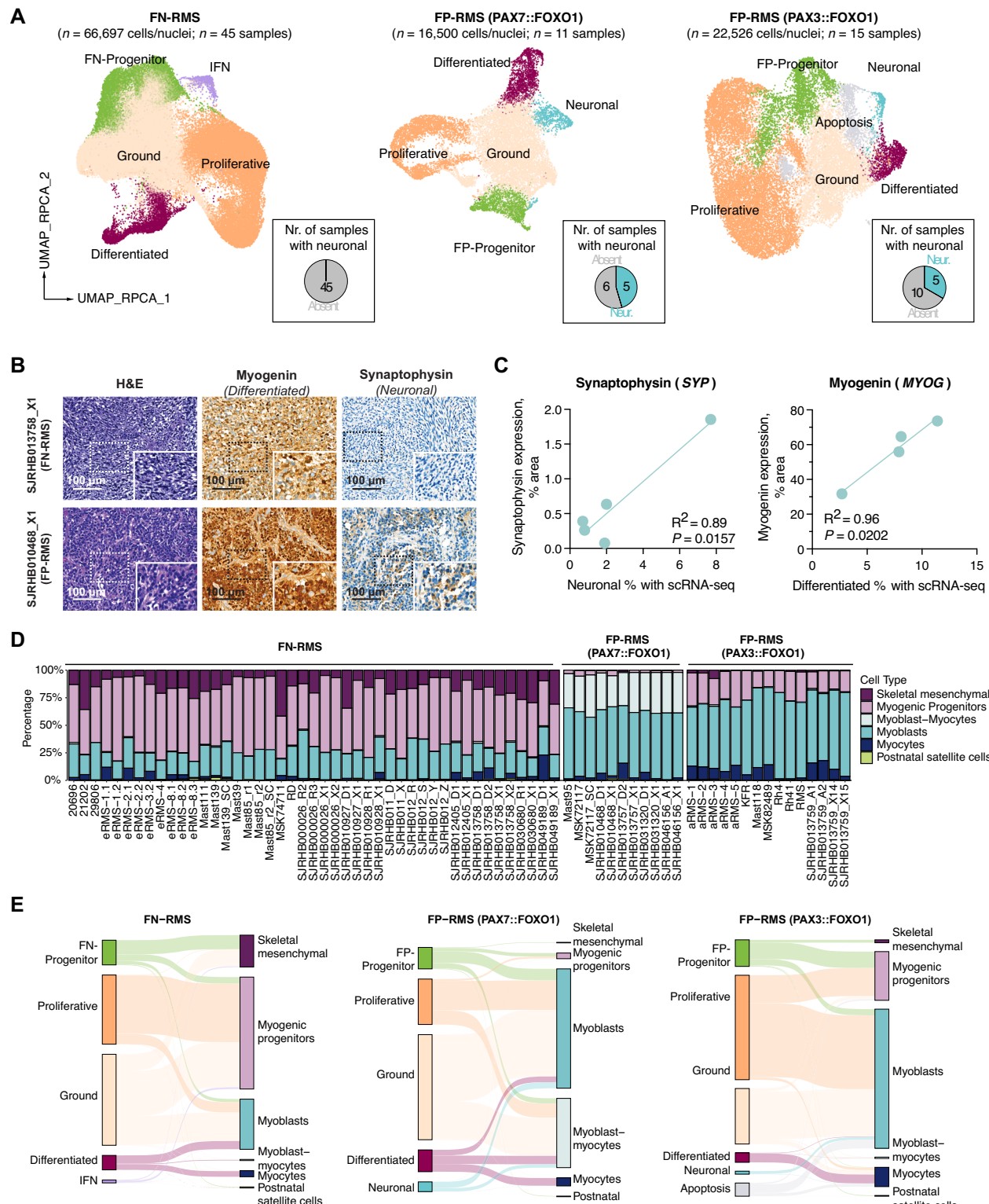

**Fig. 4 | Subtype analysis reveals shared RMS cell heterogeneity with human skeletal muscle development and tumor-derived cells states. A** UMAP plots of FN-RMS, PAX7::FOXO1, and PAX3::FOXO1 FP-RMS. Cells/nuclei were integrated independently and colored based on cell state. The number of samples having ≥1% neuronal cells are shown in the pie charts at the bottom right of each graph. **B** Immunohistochemistry staining of O-PDX samples stained for myogenin (*MYOG*, marker of the muscle differentiated subpopulation) and synaptophysin (*SYP*, marker of the neuronal subpopulation). Staining is representative of $n = 4$ tested samples for myogenin and $n = 5$ tested samples for synaptophysin. **C** Correlation between the proportion of neuronal or differentiated cells identified by sc/snRNA-

seq and immunohistochemistry for synaptophysin and myogenin, respectively ($n = 5$ or $n = 4$ FP-RMS PDXs, respectively). The coefficients of determination ($R^2$) and $P$ values of the linear regressions are shown. Source data are provided as a Source Data file. **D** Comparison of RMS cell state heterogeneity with cell types found in human skeletal muscle development as defined by Xi et al. 2020[33]. Cell types from human skeletal muscle development were projected onto RMS cells using an unbiased cell-type prediction analysis. **E** Sankey plots showing the proportion of tumor cells classified according to their most similar human developmental equivalent from Xi et al. 2020[33] based on unbiased cell-type prediction analysis.

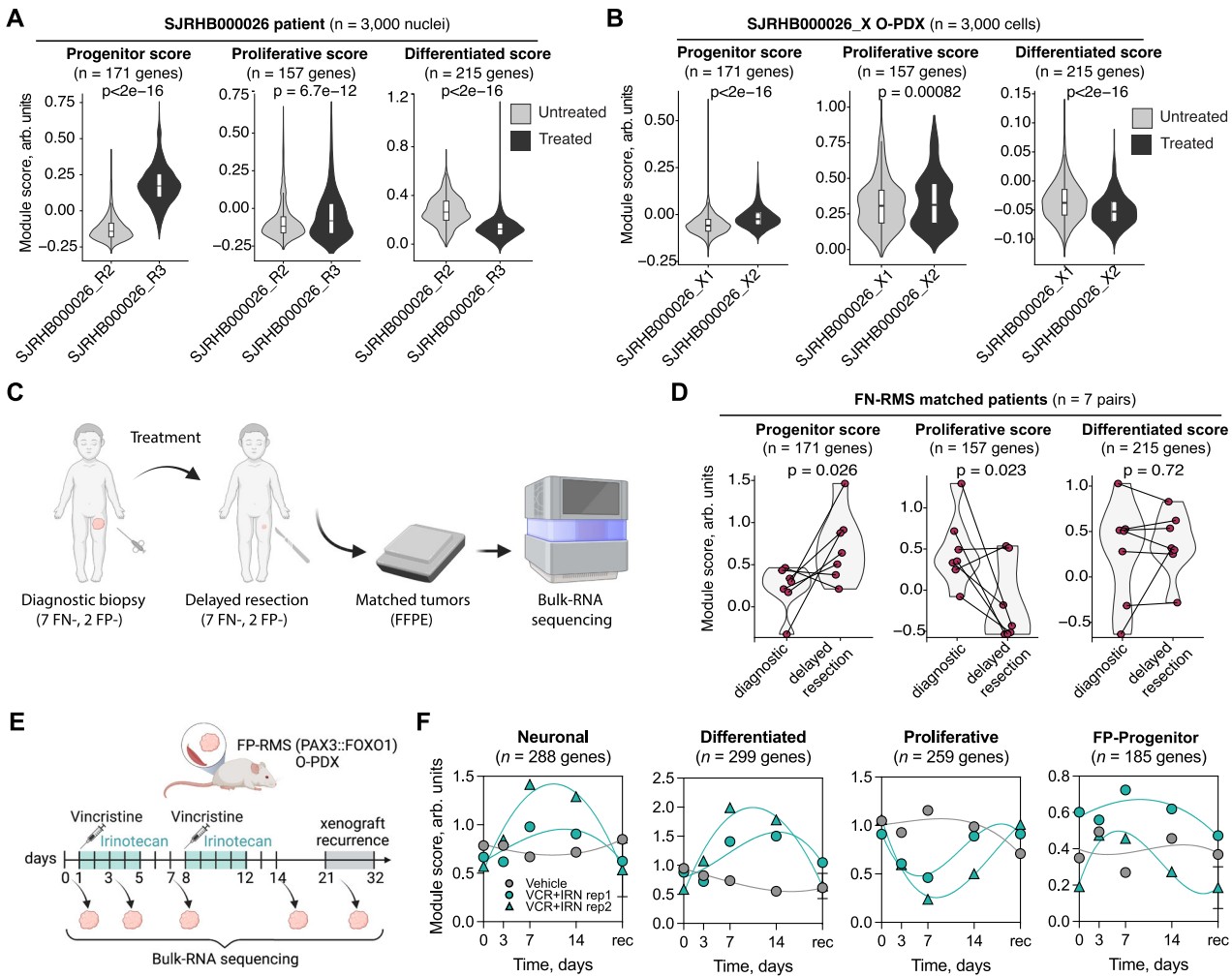

**Fig. 5 | Identification of therapy-resistant RMS metaprograms and cells states in RMS.** Metaprogram scores assigned across all cells/nuclei derived from a matched patient sample collected before (SJRHB000026_R2) and during (SJRHB000026_R3) treatment (**A**) or from PDXs derived from those patient samples (**B**). Boxplots denote Tukey's whiskers (25–75 percentile represented by minima-maxima; statistical median as center). Two-sided student's *t* test with *p* values noted in the figure. **C** Paired samples obtained from patients who underwent a pre-treatment biopsy and a delayed resection amidst therapy were processed using bulk RNA-sequencing. Created with BioRender.com. **D** Violin plots

showing metaprogram scores calculated using the RMS-atlas signature gene sets for 7 matched pairs of FN-RMS samples. Two-sided student's *t* test with *p* values noted in the figure. **E** Longitudinal biopsies from mice bearing a FP-RMS orthotopic PDX, SJRHB013759_X14, which were treated with vehicle or vincristine+irinotecan (VCR + IRN). Five longitudinal samples were obtained from each mouse, and samples underwent bulk RNA-sequencing. Created with BioRender.com. **F** Metaprogram scores of control treated mouse or those treated with VCR + IRN. Scores were calculated using signature gene sets from PAX3::FOXO1 FP-RMS (Supplementary Data 5). Source data are provided as a Source Data file.

Due to the rarity of FP-RMS, our matched cohort had few evaluable samples (*n* = 2; Fig. S6C), which limited our ability to investigate treatment-induced shifts specifically within FP-RMS. To overcome this limitation, we performed bulk RNA-sequencing of tissue obtained from longitudinal biopsies of a FP-RMS orthotopic PDX, SJRHB013759_X14[24]. Xenograft-bearing mice were either treated with vehicle or with chemotherapy (vincristine+irinotecan), and sedated needle biopsies were obtained at 5 time points during therapy: day 0 (pre-treatment), day 3, day 7, day 14, and at recurrence (Fig. 5E). Compared to the vehicle-treated control, tissue obtained from chemotherapy-treated tumors showed upregulation of the differentiated and neuronal signatures at early time points and a return to basal levels at recurrence (Fig. 5F). Proliferative scores were downregulated at early time points and returned to basal levels at recurrence consistent with the anti-proliferative properties of chemotherapy (Fig. 5F). We also observed that Progenitor and Differentiated muscle scores showed a trend of enrichment in delayed resection, whereas Proliferative score was lower in two matched FP-RMS bulk RNA-sequencing. In total, these studies identify important

cell states that are retained and expanded after therapy in both FN- and FP-RMS.

## Discussion

Single-cell sequencing technologies have provided unprecedented insight into the intratumoral heterogeneity of a variety of cancers. However, the application of this technology to rare pediatric cancers has been limited by tissue availability, cost, and standardization of bioinformatic analyses. For example, there are approximately 350 new diagnoses of pediatric RMS in the United States annually[2], which limits the ability of any one institute to accrue a sizeable cohort of samples. Here, we combine datasets from three independent studies to define distinct tumor cell heterogeneity and malignant cell states shared with human muscle cells. This consensus analysis also provides a framework for multi-investigator cooperation that we hope will be an example for future efforts to better understand rare and understudied tumors. Importantly, this work uncovered unexpected biology related to transitional and therapy-resistant cells states and challenge findings across our and others' previous work, which could only be

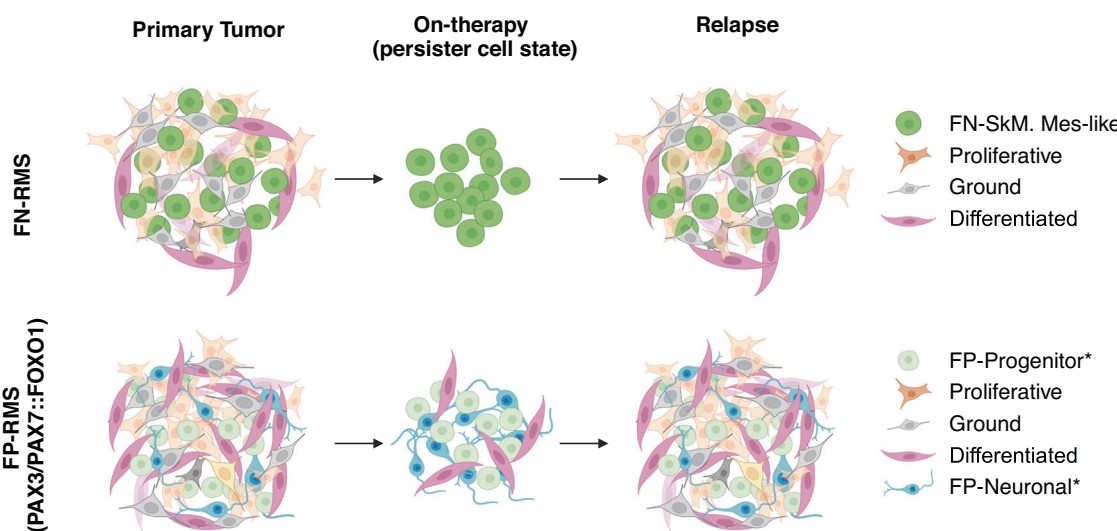

**Fig. 6 | Graphical summary.** Proposed model of persister cells during treatment that contribute to relapse in FN-RMS (top) and FP-RMS (bottom). Asterisks denote tumor acquired cells states. Image created with BioRender.com.

accomplished with a consensus view of our data and cross-institutional cooperation.

In total, we analyzed 72 samples including 27 FP-RMS, 43 FN-RMS, and 2 MYOD1[L122R] mutant SS-RMS samples. We identified shared cell states across RMS samples, which we named based on expression similarity shared with human skeletal muscle development including: (1) progenitor cells that express mesenchymal markers and we now name "FN-skeletal muscle mesenchymal-like" (FN-SkM.Mes-like) and "FP-progenitor" cells; (2) differentiated cells that express differentiated muscle-lineage markers; (3) proliferative cells that are enriched for expression of cell cycle genes and largely fail to express progenitor/mesenchymal genes or differentiated muscle genes; and, (4) ground state cells that do not show enrichment of any cell state markers. Our new analysis also identified previously underappreciated transitional cell states in both FN- and FP-RMS, suggesting a continuum of states across RMS. Finally, we identify that a subset of FP-RMS harbor small numbers of cells that express neuronal genes referred to as "FP-neuronal" cells, and that these cells are enriched during chemotherapy. In total, this work provides a standardized nomenclature for RMS cell subpopulations, introduces a transcriptomic muscle-lineage score for assessing cell state, provides cell state signature profiles for harmonizing future studies of RMS heterogeneity, and confirmed the existence of a tumor-acquired neuronal cell state in a subset of FP-RMS (Fig. 6). While the 72 datasets used to generate this meta-analysis were exclusively generated from droplet-based 3′-biased sequencing methods, we demonstrate that our signatures are broadly applicable to other single-cell sequencing technologies by validating them against a cohort of 19 samples generated from a plate-based method, SORT-seq[23]. One limitation of the defined RMS atlas is that RNA sequencing largely contains 3′ enriched sequences and provides limited full-length gene coverage. Thus, we were unable to map somatic mutations within individual cells and assess impacts on overall cell state. We anticipate that new and emerging techniques, such as long-read sequencing of droplet-based single-cell libraries[39,40] or combined single-cell DNA/RNA sequencing[41,42], will be powerful tools to simultaneously interrogate clonal heterogeneity and cell state heterogeneity.

Our cohort included a diversity of datasets generated from patient tumors, PDXs, primary cultures, and commercially available cell lines. Consistent with earlier reports[21,22,24], we note that PDXs maintain the underlying heterogeneity of patient tumors. PDXs expanded and processed at two different institutions present similar diversity of cell subpopulations (Fig. S1D), indicating that PDXs represent a reproducible experimental model for studying RMS heterogeneity. In contrast, both primary and commercial cell lines were enriched for the most proliferative compartment of RMS tumors and were depleted of FP-neuronal cells (Fig. S3C), suggesting that caution must be applied in using cell lines to model therapy response. The emergence of 3-dimensional culture models of RMS[23,43,44] may present a potential "middle ground" for in vitro models that may more faithfully recapitulate the underlying heterogeneity of RMS. Indeed, recent work from DeMartino et al. indicate that organoids preserve the malignant cell states of RMS, with absence of non-malignant cells of the tumor microenvironment[23]. Genetically-engineered models represent an alternate approach to model RMS, and multiple genetic models of RMS and have been generated in mice and zebrafish[45–50]. It remains an open question of to what degree these engineered models mimic the heterogeneity of human RMS. We anticipate that the expression signatures and lineage score generated within this study will be applicable as a future tool for comparing genetically engineered models of RMS to that of patient samples.

Our findings also contribute to a growing body of literature describing rare cancer cells with the capacity to propagate and re-establish tumors after therapy, which have sometimes been called cancer stem cells or tumor-propagating cells[51–53]. We identified a group of cells expressing a progenitor signature including markers of early muscle progenitors such as *CD44*, *EGFR*, and *THY1* (CD90). Previous work using flow sorting for cell surface markers such as CD133, CD44, and EGFR have validated the existence of these cells in FN-RMS and demonstrated that they propagate FN-RMS both in vitro and when grown in immunocompromised mice[21,24,54–58]. Intriguingly, 'Progenitor' FN-RMS cells shared gene expression similarity to a newly defined Skeletal muscle mesenchymal cell state that has bipotent capability to make muscle and osteogenic lineage cells[33]. Based on functional studies showing that this RMS cell state can drive tumor growth after stress and has the capacity to make osteogenic lineage cells[21], we have refined our naming of this cell state as "FN-SkM.Mesen-like". Our analysis also suggested that FN-RMS replicate the broad diversity of fetal muscle development cell states and have a shared developmental hierarchy with early developing fetal muscle found after 7 weeks post-conception. These findings contrast with Patel et al. which proposed that FN-RMS recapitulate an earlier mesodermal specification program in mice[24]. This difference is likely attributable to interspecies variation in myogenesis, especially since bipotent SkM.Mesen cells have yet to

be identified in mice, or that comparison with human muscle development did not include cell types from the earliest stages of mesodermal specification that begin at 24 days post-conception in humans. Finally, our data suggests that FN-SkM.Mesen-like cells are largely quiescent and are likely the therapy persistent cells that re-establish tumors after treatment. Indeed, a similar phenomenon where cells with characteristics of progenitors from the hematopoietic, colon, and brain lineages have been proposed to play roles in leukemia, colorectal cancer, and glioblastoma, respectively[59–61].

Our analysis also uncovered that FP-RMS do not display the same rigid developmental hierarchies as found in normal development and may contain different therapy persister cell states. For example, FP-RMS tumors have fewer overall proportions of progenitor cells with some tumors seemingly lacking this cell state completely. Moreover, although the FP-progenitor cells do expresses mesenchymal markers, they are transcriptionally distinct from the SkM.Mesen cells found in fetal muscle development and the FN-SkM.Mes-like state discovered here. Indeed, DeMartino et al., also independently identified key differences in mesenchymal-pathway enriched cell states in comparing scRNA sequencing expression of FP- and FN-RMS[23]. In addition, a subset of FP-RMS have tumor cells that have neuronal-pathway activation that are not found in human muscle development and yet are enriched after chemotherapy. The existence of this FP-RMS cell state is supported by immunohistochemical studies of 42 FP-RMS tumors that identified a subset of FP-RMS express marker genes including chromogranin, CD56, and synaptophysin[62]. While RMS have demonstrated histological resemblance to cells of the myogenic lineage[2,20,63], our work suggests that FP-RMS are able to transition to cell states not found in normal myogenic development. Intriguingly, lineage plasticity and neuroendocrine transdifferentiation have been reported as resistance mechanisms in multiple adult cancer types, including melanoma and castration-resistant prostate cancer[64–66]. Indeed, earlier experiments using limiting dilution cell transplantation assays into immune deficient mice showed that FP-RMS have a high frequency of tumor initiation, raising the possibility that most if not all FP-RMS cells can acquire the ability to propagate tumors in vivo[67]. The frequency by which this tumor-acquired cell states are found in FP-RMS and defining their possible role in driving therapy resistance will clearly be a major research focus for the field. Moreover, future work is needed to clarify the relationship between the myogenic and neuronal cell states, and the mechanism by which FP-RMS tumors adopt the neuronal state.

In total, our work has defined cell state heterogeneity in RMS including identifying a continuum of progenitor and muscle differentiation gene expression, two FP-RMS cell states that are not shared in muscle development, and subtype specific, therapy-persistent cell states that likely drive tumor regrowth at relapse.

## Methods

### Human subjects and animal experiments

De-identified human tumor tissue for this study were obtained and processed after approval by the St. Jude Institutional Review Board. In particular, formalin-fixed tissues from diagnostic and on-treatment RMS tumors were obtained as part of the RMS13 trial at St. Jude Children's Research Hospital (NCT01871766) for the analysis performed in Fig. 5C, D.

### Animal experiments

Frozen needle biopsy tissue from orthotopic patient xenograft tissue for Fig. 5 were obtained after approval for all procedures and handling by the St. Jude Institutional Animal Care Use Committee (IACUC). Biopsied tissue were generated as part of an earlier study[24]. Immunodeficient mice were housed according to IACUC standards using barrier conditions and isolation cages to minimize pathogen exposure. The housing facility operates with an alternating light schedule (12 h on, 12 h off) and has a dedicated isolated ventilation system. All mice were fed and provided water ad libitum.

### scRNAseq/snRNA analysis

**Public datasets.** The 10X Genomics scRNAseq/snRNAseq data was collected from previously published datasets[21,22,24,25]. All datasets are available at the NCBI Gene Expression Omnibus (GEO) database under the following accession numbers: GEO: GSE218974 (Danielli et al.; $n = 17$ samples), GEO: GSE195709 (Wei et al.; $n = 18$ samples), GEO: GSE174376 (Patel et al.; $n = 36$ samples), GEO: GSE113660 (Cheng et al.; $n = 1$ sample).

**Data pre-processing.** For samples derived from Wei et al. and Danielli et al., filtered Seurat objects were downloaded from GEO repositories. Those objects were generated as previously described[21,22] using the 10X Genomics Cell Ranger pipeline (version 3.0.1 in Danielli et al.; version 3.1.0 in Wei et al.) to map raw sequencing FASTQ files to the human genome reference (hg19 for patient samples, hg38 for primary cultures) or to both the human hg19 and mouse mm10 references (for PDX samples). Low-quality cells, defined as cells with high mitochondrial ratio (>15% in Danielli et al., >20% in Wei et al.), low expressed gene number (<200 in Danielli et al., <1000 in Wei et al.), high expressed gene number (>8000), and PDX cells potentially derived from mice (mouse reads ratio >5% in Wei et al.) were already filtered out.

For samples derived from Patel et al., we generated single-cell Seurat objects following the original pipeline[24]. In short, raw sequencing FASTQ files available from GEO were aligned to the human hg19 (for patient samples) or to the combined human hg19 and mouse mm10 references (for PDX samples) using the 10X Genomics Cell Ranger pipeline (version 3.0.0). Low-quality cells, defined as cells with high mitochondrial ratio (>10%), low (<400) or high expressed gene number (>7000), were filtered out.

In their original studies, Wei et al. and Patel et al. both used inference of copy-number alteration analysis on the patient-derived datasets to differentiate between malignant cells (i.e., cells/nuclei harbor tumor-specific copy-number alterations) and non-malignant cells. In this study, we included only single-cell/nucleus profiles that were annotated as malignant in their respective original papers.

For the cell line Rh41, we downloaded the filtered gene-cell matrix available on GEO, that was generated as previously described[25] using the 10X Genomics Cell Ranger pipeline (version 2.0.1) to map raw sequencing FASTQ files to the human hg38 genome reference. Low-quality cells, defined as cells with high mitochondrial ratio (>15%), low (<200) or high expressed gene number (>8000) were filtered out.

**Merging of single-cell transcriptome data.** To create the RMS atlas, we first subset each sample to typically $n = 1500$ randomly selected cells (Supplementary Data 1), and then merged raw count matrices using Seurat's *merge* function. This resulted in a total of $n = 107,523$ cells from $n = 72$ RMS samples (Supplementary Data 1).

To create the three subtype-specific RMS atlases [(1): FN-RMS ($n = 45$ samples); (2): PAX3::FOXO1 FP-RMS ($n = 15$ samples); (3): PAX7::FOXO1 FP-RMS ($n = 11$ samples)], we merged subtype-specific raw count matrices using Seurat's *merge* function.

**Normalization and data reduction.** After merging, we log-normalized the data, selected the top 2000 variable features for downstream analyses, and scaled the gene expression. We then performed principal component analysis (PCA) and, based on elbow plot, selected the top $n = 15$ principal components (PCs) to consider for downstream analysis. To visualize the cells, we reduced the dimensionality of the datasets using Uniform Manifold Approximation and Projection (UMAP).

**Batch correction and clustering.** To remove the batch effects from different samples, we integrated the datasets following Seurat's integration pipeline (https://satijalab.org/seurat/archive/v3.0/integration.html), which is based on the identification of anchor cells between pairs of datasets. We first normalized and selected $n = 2000$ variable features for downstream integration from each dataset. We then scaled the data and ran PCA on each object. We identified anchors using reciprocal PCA (RPCA), the suggested option for large datasets, and integrated the datasets using the *IntegrateData* function. We then scaled and centered the gene expression, performed PCA. Based on elbow plot, we then selected the number of PCs to retain for downstream analyses. We built a K-nearest neighbor (KNN) graph, used the Louvain algorithm for clustering the cells (resolution of 0.2–0.3), and visualized the cells using UMAP plots. The number of identified clusters stabilizes at a resolution of around 0.3, yielding a total of 12 clusters. For this reason, we performed our analyses using a resolution of 0.3. To identify genes that were enriched within each cluster, we used Seurat's *FindAllMarkers* function filtering for genes with fold-change $>\log_2(0.25)$ in the subtype-specific datasets and $>\log_2(0.3)$ in the integrated dataset, and expressed in at least 25% of cells in the cluster.

**Annotation of cell clusters.** After clustering, we assigned cell states based on the expression of known markers and gene set enrichment analysis[68–70]. Specifically, we used the marker genes of each cluster as input for Enrichr (https://maayanlab.cloud/Enrichr/)[68–70], and looked at the GO Biological Process 2023 enriched terms. To annotate and collapse the clusters that contained similar lineages, we used the expression of known markers and gene set enrichment analysis[68–70]. For example, clusters 6 and 9 of Fig. 1D both expressed high levels of the muscle differentiation markers *MYOG, MYL4, MYH3*, and were therefore collapsed into one category ('Differentiated'); clusters 8 and 1 both expressed high levels of the collagen and extracellular matrix genes *COL3A1, COL1A1, FN1*, and were therefore collapsed into one category ('Progenitor').

**RMS cell scoring for meta-programs**
**Cell-state specific module scoring.** To score each cell based on previously identified metaprograms, we selected the gene markers of the original publications as gene inputs (ref. Supplementary Data 4). We then assigned cell state-specific module scores using the *AddModuleScore* Seurat's function. This function works by taking an input set of genes and comparing their average relative expression to that of a control set of $n = 100$ genes randomly sampled[71].

To calculate the consensus *progenitor*, *proliferative*, and *differentiated* marker gene set, we selected cell state markers that were enriched in at least two original publications[21,22,24] (Supplementary Data 4), or in one of the original publications and in the integrated RMS atlas clusters. We then assigned cell state-specific module scores using the *AddModuleScore* Seurat's function, as described above.

To score each RMS cell (i) along a continuum of myogenic differentiation, we defined the *muscle lineage score* ($MLS_i$). We calculated the $MLS_i$ for each cell by subtracting the *progenitor* ($P_i$) score from the *differentiated* ($D_i$) score. Unless otherwise specified, cells were scored using the new consensus *progenitor*, *proliferative*, and *differentiated* markers. The scores were scaled using the *ScaleData* Seurat's function to center the expression values.

**Cell-cycle scoring.** After integration, we assigned cell cycle scores using Seurat's *CellCycleScoring* function, which relies on gene signatures that have been previously shown to characterize S and G2/M cell cycle phases[71]. We distinguished high cycling (S-scores or G2/M scores >0) from low cycling cells (S-scores <0 and G2/M scores <0) based on S and G2/M scores.

**Comparison of RMS tumors with single-cell reference data from human development.** To infer comparisons between RMS tumors and human skeletal muscle development, we re-analyzed a scRNAseq dataset of human skeletal muscle development (GEO: GSE147457)[33]. We downloaded gene expression matrices and their corresponding metadata information for the myogenic subsets derived from embryonic development (1), fetal development (2), juvenile (3) and adult (4) directly from the authors (http://cells.ucsc.edu/?ds=skeletal-muscle). After merging the raw count matrices of the individual datasets, we log-normalized the data, selected the top 2000 variable features for downstream analyses, and scaled the gene expression. We then performed PCA and, based on elbow plot, selected the top $n = 10$ PCs for downstream analysis. To visualize the cells, we reduced the dimensionality of the datasets using Uniform Manifold Approximation and Projection (UMAP).

To recognize the cell types and developmental time points at which RMS tumors might arise, we used SingleR[38], a computational framework that takes a dataset with known labels as an input and that transfers them onto a test dataset based on similarity to the reference. Specifically, we projected signatures from the human development dataset[33] onto our combined FN-RMS, PAX3::FOXO1 FP-RMS and PAX7::FOXO1 FP-RMS single cell objects.

**scCODA.** Bayesian compositional modeling was used to perform differential populations testing using scCODA with default parameters[31]. A false discovery rate of 0.05 was used as a cutoff between statistically credible and non-credible differences.

**Bulk RNA-seq**
**Sample collection.** Flash-frozen orthotopic PDX biopsy tissue generated from ref. 24. underwent RNA isolation using Trizol (Invitrogen) extraction, as per manufacturer instructions. Samples were manually homogenized in 800 µl Trizol within microcentrifuge tubes using a disposable plastic pestle (Fisher Scientific). An additional 200 µl of chloroform was added and mixed via inverting for 3 min at room temperature. Samples were then centrifuged at 12,000 x $g$ for 15 min at 4 °C. The aqueous layer was transferred to a new microcentrifuge tube, and RNA was precipitated by the addition of an equal volume (approximately 500 µl) of isopropanol and 1 µl glycogen (Thermo Scientific). Samples were incubated at room temperature for 10 min, followed by centrifugation at 12,000 x $g$ for 15 min at 4 °C. Pellets were washed twice with 75% ethanol and resuspended in nuclease-free water. RNA quality was estimated using RNA ScreenTape on a TapeStation automated electrophoresis instrument (Agilent). Sequencing libraries were generated using the TruSeq Total Stranded RNA Library kit (Illumina) using 250–1000 ng of input RNA. Libraries underwent 100 nucleotide paired end sequencing on a NovaSeq 6000 (Illumina).

For extracting RNA from formalin-fixed paraffin-embedded (FFPE) tissue of patient tumors, 5 µm tissue scrolls were processed using the Maxwell RSC RNA FFPE instrument (Promega). RNA concentration and quality was determined using a TapeStation automated electrophoresis instrument (Agilent). Samples with a $DV_{200}$ score (calculated as a percentage of nucleic acid fragments >200 nucleotides) above 20% were used for downstream library generation using the SMARTer Stranded Total RNA – Pico RNA-seq kit v2 (Takara). Libraries were sequenced using 100 nucleotide paired-end sequencing on a NovaSeq 6000 (Illumina).

**Data pre-processing.** Following sequencing, all RNA-seq libraries were processed using an automated computational pipeline. Briefly, sequenced reads were trimmed and underwent quality control using FastQC, followed by aligning and counting using STAR[72]. To generate additional alignment metrics, the STAR-aligned BAM file were analyzed using the 'CollectRNASeqMetrics' command from the Picard pipeline. Duplicate reads were removed using the GATK 'MarkDuplicates'

command, and then RSEM was used to generate tpm count matrices using the 'rsem-calculate-expression' command.

**Bulk-RNAseq scoring.** To score FFPE tissues and orthotopic PDX biopsies for the *progenitor*, *proliferative* or *differentiated* scores, we first created a Seurat object using the already TPM-normalized read count matrix, and log-normalized the count matrix expression values + 1. We then scored individual samples for the cell state-specific module scores using the *AddModuleScore* Seurat's function. We plotted the scores after scaling and centering the expression values using the *ScaleData* Seurat's function.

## Immunohistochemistry
Tissues were fixed in 10% neutral buffered formalin, paraffin embedded, sectioned at 4 µm, and mounted onto glass slides (Superfrost Plus; 12-550-15, Thermo Fisher Scientific, Waltham, MA). Slides were then dried for 20 min at 60 °C, deparaffinized, and stained with hematoxylin and eosin (Richard-Allan Scientific) or used in immunohistochemistry experiments. HE sections were stained and coverslipped using the HistoCore SPECTRA Workstation (Lecia Biosystems). Serial sections were immunolabeled with Synaptophysin (Abcam, ab32127, 1:400) using a Ventana Discovery Ultra autostainer (Roche, Indianapolis, IN) and the following conditions: Heat-induced epitope retrieval, Cell Conditioning Solution ULTRA CC1 (950-224, Roche) for 32 min and visualization with DISCOVERY OmniMap anti-Rb HRP (760-4311, Roche), Hematoxylin II (790-2208, Roche), and Bluing reagent (760-2021, Roche). MYOGENIN staining (Abcam, ab1835, 1:150) was performed on a Ventana Discovery Ultra autostainer (Roche, Indianapolis, IN) using the following conditions: Heat-induced epitope retrieval, Cell Conditioning Solution ULTRA CC2 (950-223, Roche) for 60 min and visualization with DISCOVERY OmniMap anti-Rb HRP (760-4311, Roche), Hematoxylin II (790-2208, Roche), and Bluing reagent (760-2021, Roche). Whole slide images to a 20x scalable magnification were created using a PANNORAMIC 250 Flash III digital slide scanner (3DHISTECH Ltd, Budapest, Hungary). Images were taken using the HALO v3.6.4134.137 software program (Indica Labs) and analyzed using HALO v3.2.1851.354 and the Area Quantification FL v2.3 algorithm to determine the area of immunoreactivity for each marker (all Indica Labs, Albuquerque, NM). Visual interpretations of immunohistochemical staining were conducted by a board-certified veterinary pathologist and in a manner that was blinded to the experimental condition of each mouse and compared with image analysis findings.

## Statistics and reproducibility
No data were excluded from the analyses. The experiments in this study were not randomized. All box plots in this study are reported as using the Tukey method, with bars extending from the 25th to $75_{th}$ percentiles and center bar denoting the statistical median. Relevant statistical testing for differential expression analyses and two-way comparison are reported in the text and figure legend.

## Reporting summary
Further information on research design is available in the Nature Portfolio Reporting Summary linked to this article.

## Data availability
Published single-cell/nucleus RNA-sequencing data were obtained from the Gene Expression Omnibus (GEO): Danielli et al. (GSE218974); Patel et al. (GSE174376); Wei, et al. (GSE195709); Cheng et al. (GSE113660). RNA-sequencing data generated from matched patient samples before and during therapy as well as biopsied FP-RMS orthotopic PDXs are available at the GEO under accession number GSE240287 and GSE240308, respectively. The RMS single-cell objects generated in this study have been uploaded on FigShare [https://figshare.com/projects/RMS_consensus_analysis/194417]. Source data

are provided as a Source Data file. Source data are provided with this paper.

## Code availability
The code used to generate the results reported in this manuscript are available through a Github repository [https://github.com/Sara-Danielli/RMS-metadata].

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

## Acknowledgements
This work was funded by the Sarcoma Foundation of America (2022 SFA 13-22, B.W.S. and S.G.D.), the Hyundai Hope on Wheels Foundation (A.G.P.), the Damon Runyon Cancer Foundation (#DRSG-33P-20, A.G.P.). the Alex's Lemonade Stand Foundation (M.A.D. and A.G.P.), CureSearch (D.M.L.), American Lebanese Syrian Associated Charities (M.A.D. and A.G.P.), the Friends of TJ and Summer's Way Foundation (Y.W.), MGH ECOR Medical Discovery Award (Y.W.), the Rally Foundation (D.M.L.), Infinite Love for Kids Fighting Cancer (D.M.L.), and the NCI K99CA278696 (Y.W.), R01CA276116 (D.M.L.), R01CA269213 (D.M.L.), R01CA226926 (D.M.L) U54CA231630 (D.M.L.), the Childhood Cancer Research Foundation Switzerland (B.W.S.), and the V-foundation (D.M.L.). We thank the St. Jude Clinical Biomarkers Laboratory for assistance with RNA extraction of formalin-fixed paraffin embedded tissues, the St. Jude Comparative Pathology Core for assistance with immunohistochemical staining of samples, the St. Jude Hartwell Center for Biotechnology for sequencing support, and the St. Jude Center for Applied Bioinformatics for support with RNA-sequencing analysis. Several figures were created with BioRender.com (Figs. 1A, 5C, E and 6).

## Author contributions
D.M.L., A.G.P., M.W., and B.W.S supervised the study design and writing. S.G.D. performed the analysis and generated figures. S.G.D. and Y. W. coordinated the collaboration, provided feedback on ideas and figures, and wrote the manuscript. M.A.D. and E.S. provided samples for the analysis of treatment-induced expression signatures. H.S. supervised immunohistochemical staining and performed analysis of histologic data.

## Competing interests
The authors declare no competing interests.
