## [Peer Review File · Nature Communications]

REVIEWER COMMENTS

Reviewer #1 (Remarks to the Author):

The authors report a harmonized data analysis of 72 datasets from primary tumors, PDXs, and cell models derived from rhabdomyosarcoma (RMS) patients. The consolidation and harmonization of these data is an important contribution because of challenges in comparisons of datasets across labs and through the use of different analysis pipelines, which can create apparent differences in expression data. Through their approach, the study reveals 4 major classes of RMS sub-populations, including progenitors, proliferative, differentiated and ground state cells. The study builds upon exciting findings from (PMID: 35982179, PMID: 35483358, PMID: 37244912, and PMID: 36753540). This is impactful because the standard of care therapy regimen has not changed substantially in decades, and understanding how distinct sub-populations of RMS cells respond to therapy will reveal key mechanisms.

The authors identify “significant overlap” between the published gene signatures that were revealed in the previously reported (PMID: 35982179, PMID: 35483358, PMID: 37244912, and PMID: 36753540) studies which adds significance the authors’ findings of the major cell populations within all RMS tumors. Two key exceptions to the generality of the major clusters or sub-populations of cells existing in all RMS tumors, irrespective of subtype, was the “interferon cluster” found in FN-RMS and the neuronal cell cluster existing in FP-RMS tumors. Importantly, the authors identify that neuronal or differentiated cell sub-populations are selected for after standard of care therapy.

The impact of the manuscript is high, as is the significance to the field. However, despite the overall enthusiasm for the study, and the harmonization of data across labs to generate a molecular portrait of rhabdomyosarcoma, there are fundamental mechanism questions that remain unaddressed in the work. Specifically, to what extent is expression of the major alterations (associated with the distinct genetic subtypes in the study) required for tumor maintenance or therapy resistance within distinct cell populations in RMS? Similarly in FN-RMS, are the major driving mutants expressed across cell populations, or specifically within therapy-resistant cells? The meta-analysis and reanalysis of 72 datasets is a major effort and key advance in the field, but the study would be strengthened from further use of these data to address mechanism.

Major Comments:

1) It is unclear what new data is being generated by this study in the main text. To consolidate and harmonize data analyses is critical for generating meaningful conclusions from data, but it would strengthen the study to also introduce new datasets as well for validation and expansion of the findings.

2) It is not very clear if the Louvain clusters in Figure 1E corresponds to diagnosis or pathology, or subtypes. If not, then it begs the questions on if the clustering/visualization is significant. If yes, please modify the figures to make it clearer.

3) The finding that 4 major subpopulations exist within all RMS tumors is interesting. “Lineage score” should be clearly defined in the Methods section. As it stands right now, it is not clear if the score is normalized and what it represents. The tumors which display low lineage scores should be further investigated and characterized. This is especially significant with the MYOD-mutant (MSK74711) sample in this study, which may display transcriptional features of other lineages classes. It would strengthen the study to more deeply characterize and determine what these alternative expression signatures are, especially for ERMS samples with lower lineage scores.

4) It would strengthen the manuscript to clarify whether the unique presence of the neuronal-profile cell clusters in FP-RMS tumors (but not in cell lines) were the result of anatomic location and/or resection versus an intrinsic property of the FP-RMS cells. Is it possible to verify that expected RMS alterations are in the same cell populations that are classified as neuronal? Addressing these questions could reveal key mechanisms, and overcome a limitation of the study. Furthermore, it is unclear which samples were determined to have the neuronal signature. Table S7 only lists the neuronal percentage per sample, but it is not evident where the cut-off is—perhaps the samples called as containing a significant neuronal percentage should be highlighted. Furthermore, Table S6 shows that the neuronal cluster has 3/10 or 4/10 significant neural terms in the GSEA, and muscle-related terms are just as/more common. Maybe this should be noted in the text, to emphasize that these are cells with multiple significant gene expression patterns.

5) In FN-RMS, is the expression of RAS-mutants or MYOD-mutants found across clusters (e.g., progenitor, proliferative, differentiated, ground state), or are driver alleles only expressed in select populations of cells before or after therapy?

6) There is no indication that the code for generating the analysis is publicly available; this should be added to the manuscript. However, the methods clearly explain the analysis and should be reproducible for anyone familiar with scRNA-seq. The authors are commended for providing the extensive supplementary tables listing gene signatures for various clusters, etc to provide a resource for other labs. Please include a code repository for the analysis in the revision.

Minor Comments

1) Reference #21 is a Biorxiv pre-print and should be updated to reflect the most recent form of that work.

2) A more in depth characterization of “transiting-differentiated” cells would strengthen the manuscript. What does “had lower levels of ... cells states genes” mean? Should also correct the typo here.

3) The section heading “The muscle lineage score stratifies between RMS subtypes” anthropomorphizes and should be revised. Perhaps an alternative like “The muscle lineage score reveals key distinctions between RMS subtypes” would be more clear.

4) The statement in lines [219-221] “Yet, when we analyzed the overlap in expressed genes from these seemingly shared states, we observed that progenitor cell states were not transcriptionally the same across each tumor subtype (Figure S3B)...” is unclear as written. Does this mean “across” subtypes as in comparing ERMS to ARMS, or across samples within a genetically defined subtype? Revision to clarify would strengthen them manuscript.

5) On line 446, a reference needs to be filled in.

6) The model figure, Figure 6, is very clear and communicates the main points of the manuscript well. However, the differentiated population, which has an increasing signature early with therapy in FP-RMS according to Figure 5, is not indicated in this population in FP-RMS. Only the neuronal and FP-progenitors are shown. This model figure should be updated so it fully reflects the conclusions of the manuscript.

7) In lines 311-312, the authors state “these studies identify important new cell states that are retained and expanded after therapy.” It is important to note that bulk RNA-seq is used in Figure 5, and so the data does not necessarily indicate the proportion of cells of each type. This should be clarified in the text to aid the reader, given that Figure 6 assumes the proportion of each population has been determined on-therapy and at relapse.

Reviewer #2 (Remarks to the Author):

The team provides a nice view of RMS expression by developing a single cell atlas across primary tumors and cell lines. The dataset is a tour de force especially because there are were many unique patient samples. They define four populations: progenitors, proliferative, differentiated, and ground cells. They stratify these RMS cells along the continuum of human muscle development and show that some RMS cells share expression patterns with fetal/embryonal myogenic precursors rather than postnatal satellite cells. They also identify populations resistant after chemotherapy and new states in the RMS cells not in development. Overall, an important and interesting study for the field. Few comments/questions below:

1. FN-RMS closely resemble bipotent SkM.Mesenchymal cells-how? Based on both marker expression and function? How consistent/what are the differences in these two populations? Can the patient derived SkM.Mesenchymal cells differentiate similar to fetal developmental cells or different? What percentage of SkM.Mesenchymal cells are found in FN patient samples compared to the percentage seen in human development? How varied is this across different patient samples? In the supplement can you show the % of the different cell types present in the FN samples and variability or similarities?
2. Are there candidate targets that could provide novel molecular markers for less differentiated progenitor cells compared to differentiated muscle-like cells? Are the current therapies targeting either of these populations? Or does this dataset provide new molecular targets for treatment? I.e. could be discussed
3. There is a bipotent NMP in development that gives rise to both muscle and neuronal. Does the FP RMS or FN population/s express any of these markers at any time point? (NMPs are double positive for T and Sox2 at minimum). Do they revert back to this after treatment? Or thoughts on how this neural population arises/what it is? Or what about overlap of the RMS neuronal datasets with neural crest or neuroendocrine cells? I.e. Further evaluate what neural cells are these similar to?
4. Would be interesting to see if the entire PAX3 or PAX7 networks are activated in the FP-RMS cells with downstream single cell analysis platform or chip-seq or cut and run/tag etc.
5. Is there a way to target cells within the continuum of cancer states identified from this dataset? If so perhaps a discussion of which of these states may be best to target should be included?
6. What overlap do the mouse tumor models have with the different populations identified? ie are they good models?

6. Is the fact that you see an increase in treatment-induced selection for the FN-SkM.Mes-like progenitor state in FN- RMS a good thing or do these cells then lead to cancer relapse? Do you see an increase in FN-SkM.Mes-like cells after treatment in FN-patients +/- treatment or in multiple relapses? Can this be quantified?

7. Discussion- could any of these new populations be used as biomarkers for potential to relapse? Or that therapy treatment did/did not work well?

Reviewer #3 (Remarks to the Author):

This is a well written and constructed study of rhabdomyosarcoma single cell tumor data. The authors describe a unified analysis of each groups previously published separately generated and analyzed data. They report an interesting spectrum of gene expression patterns consistent with muscle lineage; therapy resistant progenitor states in fusion negative rhabdomyosarcoma and an interesting neuronal subtype associated with the fusion positive histology. This will be an important reference work for the field, so I strongly support the effort. However, I identified a few things that may be useful in improving the study.

1. The first step that they authors take is to generate an integrated atlas of all rhabdomyosarcoma cells. This seems like an odd first step after spending much of the introduction telling us that their interest is defining differences. As the authors know the anchor based integration may be forcing cells into groups. What happens if you overlaid cells from another embryonal histology? Would the Louvian method force those cells onto the same atlas?

2. What happens if you separate the fusion positive and fusion negative and then perform the Louvian clustering? Do you get the same answer?

3. I would like to see the clustering based on the number of genes and justification for the resolution used to create the Louvian clusters. 11 clusters may be too high of a resolution for what they are seeing. The conclusion then that there are differences between the TR-differentiated and differentiated may be overly-fit. If they are making the point that they see a difference here it would be nice to see if these groups were biologically real.

4. Figure 3 shows a muscle lineage score continuum. I like the concept but thing the definition of AU Differentiated – AU Progenitor score is too crude and may be misleading. At a minimum the gene programs that are dynamically driving this observation should be shown in Fig 3.

7. One idea is that the analysis in Figure 4 is a much more accurate representation of the heterogeneity of the dataset. Again, I worry that Fig 3 has collapsed the data to a lowest common denominator which may or may not reflect reality. I might suggest moving Fig3 to the supplement

5. They mention several times that the Patel Myoblast cells were presumably absorbed by another cluster in the current analysis. Where did they go? Are they not real?

6. Given the implications of the described interferon response cells in FN-RMS and the rare neuronal cell population in FP-RMS I would ask the authors for validation using correlative assay such as RNAish or if possible IHC.

7. The therapy resistant cells may be the most impactful finding of the study. I struggle again with why the authors collapse the data and figure into FN and FP and metaprograms. Why not have a figure that broadens the analysis of the FN patient samples and a separate one using the PDX model?

8. Does the PDX study need addition replicates to be significant? I think most people will agree with decrease in the proliferative gene set but that would not be entirely surprising. The novel finding would be the other gene programs. Would doing this experiment with a single cell assay readout not be a more conclusive story? Bulk RNA felt kind of disappointing.

RE: NCOMMS-23-51656-T

“Single cell transcriptomic profiling identifies tumor-acquired and therapy-resistant cell states in pediatric rhabdomyosarcoma”

Point-by-point response to reviewer comments

We thank the reviewers for their thorough and constructive comments. Overall, we appreciate the enthusiasm from all three reviewers, and the recognition that this study represents an impactful contribution to the fields of sarcoma biology and pediatric cancer.

The reviewers' suggestions have led us to perform additional experiments and modify the manuscript accordingly. We believe that these changes have strengthened the findings of our original manuscript. Major changes and additions include:

- 1) Analyzed additional RMS scRNA-seq dataset generated by DeMartino et al. from their manuscript entitled “Single-cell transcriptomics reveals immune suppression and cell states predictive of patient outcomes in rhabdomyosarcoma” [1]. This analysis includes an additional 19 patient-derived samples and validates our original findings, demonstrating the broad applicability of our signatures and lineage scoring approach.
- 2) Expanded our analysis of FN-RMS tumors in the context of fetal myogenesis, namely bipotent SkM.mesenchymal progenitor cells; and we also experimentally validated neuronal cells in FP-RMS tumors.
- 3) Restructured Figure 5 to better show the correlation of scRNA-seq signatures with the therapeutic implications, which are considered as most important concepts appreciated by multiple reviewers.
- 4) Constructed a github repository (<https://github.com/Sara-Danielli/RMS-metadata>) and Figshare resource (https://figshare.com/projects/RMS_consensus_analysis/194417) so that members of research community can easily replicate and expand on our findings.
- 5) Performed Bayesian compositional analysis to test the impact of clinical variates of cell type composition. Specifically, we analyzed cell type composition against molecular subgroups of RMS, including fusion-negative, fusion-positive and MYOD1^{L122R} samples.
- 6) Clarified the text to better report our findings, and expanded the discussion to cover limitations and future directions.

A detailed point-by-point response to each reviewer comment is appended below.

REVIEWER #1:

The authors report a harmonized data analysis of 72 datasets from primary tumors, PDXs, and cell models derived from rhabdomyosarcoma (RMS) patients. The consolidation and harmonization of these data is an important contribution because of challenges in comparisons of datasets across labs and through the use of different analysis pipelines, which can create apparent differences in expression data. Through their approach, the study reveals 4 major classes of RMS sub-populations, including progenitors, proliferative, differentiated and ground state cells. The study builds upon exciting findings from (PMID: 35982179, PMID: 35483358, PMID: 37244912, and PMID: 36753540). This is impactful because the standard of care therapy regimen has not changed substantially in decades, and understanding how distinct sub-populations of RMS cells respond to therapy will reveal key mechanisms.

The authors identify “significant overlap” between the published gene signatures that were revealed in the previously reported (PMID: 35982179, PMID: 35483358, PMID: 37244912, and PMID: 36753540) studies which adds significance the authors’ findings of the major cell populations within all RMS tumors. Two key exceptions to the generality of the major clusters or sub-populations of cells existing in all RMS tumors, irrespective of subtype, was the “interferon cluster” found in FN-RMS and the neuronal cell cluster existing in FP-RMS tumors. Importantly, the authors identify that neuronal or differentiated cell sub-populations are selected for after standard of care therapy.

The impact of the manuscript is high, as is the significance to the field. However, despite the overall enthusiasm for the study, and the harmonization of data across labs to generate a molecular portrait of rhabdomyosarcoma, there are fundamental mechanism questions that remain unaddressed in the work. Specifically, to what extent is expression of the major alterations (associated with the distinct genetic subtypes in the study) required for tumor maintenance or therapy resistance within distinct cell populations in RMS? Similarly in FN-RMS, are the major driving mutants expressed across cell populations, or specifically within therapy-resistant cells? The meta-analysis and reanalysis of 72 datasets is a major effort and key advance in the field, but the study would be strengthened from further use of these data to address mechanism.

Authors’ response: We appreciate the reviewer’s comments. We agree that the influence of driver mutations on the myogenic hierarchy within each molecular subtype of RMS is a critical question. We will address this topic in the reviewer response below.

Major Comments:

1) It is unclear what new data is being generated by this study in the main text. To consolidate and harmonize data analyses is critical for generating meaningful conclusions from data, but it would strengthen the study to also introduce new datasets as well for validation and expansion of the findings.

Authors' response: We thank reviewer #1 for the suggestion. As we were completing this study, a new RMS single-cell RNA-seq dataset of 19 patient-derived samples became available from DeMartino et al. [1]. We have used this as a validation dataset, which was generated using a different single-cell RNA-sequencing technique (SORT-seq, as opposed to 10x droplet-based sequencing). We applied our RMS signatures and the lineage scoring approach to this new data set. Our analysis has demonstrated that our findings are applicable to this additional dataset. We observed molecular subtypes of RMS differed in the three subpopulation signatures and in the muscle lineage score.

These results are reported in **Figure S2D**. We have additionally added language in the main text reporting these results:

*“To independently validate our results, we next analyzed an additional dataset of 19 single-cell RNA-seq datasets from RMS patients reported by DeMartino et al. that used a plate-based single-cell RNA-seq technique called SORT-seq [1]. Indeed, FN-RMS tumors from DeMartino et al. had higher overall fractions of cells with progenitor signatures, while FP-RMS tumors had elevated numbers of cells with differentiated signatures (**Figure S2D**). Moreover, our lineage scoring method validated that FN-RMS tumors consistently had lower overall muscle lineage scores, while FP-RMS tumors had elevated muscle lineage scores (**Figure S2D**). In total, our data support a model where RMS cells lie in a continuum of gene expression defined by three dominant cell states including progenitor, proliferative, and differentiated cell states while also containing subtype specific gene programs found within all RMS cells from a given tumor.” (pg. 9-10)*

2) It is not very clear if the Louvain clusters in Figure 1E corresponds to diagnosis or pathology, or subtypes. If not, then it begs the questions on if the clustering/visualization is significant. If yes, please modify the figures to make it clearer.

Authors' response: We agree that the comparison of clinical covariates with degree of heterogeneity would be a valuable addition to this study. In the revised manuscript, we have performed Bayesian compositional analysis using scCODA (scRNA-seq compositional data analysis), which has been shown to robustly model and identify statistically credible differences in cell type variation from single-cell sequencing datasets [2].

We identified differences in progenitor, TR-progenitor and TR-differentiated proportions when comparing samples between FN-RMS and FP-RMS. We have included these analyses as a new figure panel, **Figure S1E**, and added the following text within the Results section of the paper:

“We next performed single-cell compositional data analysis (scCODA) that uses Bayesian modelling to quantify differences within single-cell RNA-seq clusters across cohorts [2]. scCODA analysis identified statistically credible increases in progenitor and TR-progenitor cell fractions in FN-RMS while FP-RMS had elevated numbers of TR-differentiated cell states proportions (Figure S1E).” (pg. 7)

3) *The finding that 4 major subpopulations exist within all RMS tumors is interesting. “Lineage score” should be clearly defined in the Methods section. As it stands right now, it is not clear if the score is normalized and what it represents. The tumors which display low lineage scores should be further investigated and characterized. This is especially significant with the MYOD-mutant (MSK74711) sample in this study, which may display transcriptional features of other lineages classes. It would strengthen the study to more deeply characterize and determine what these alternative expression signatures are, especially for ERMS samples with lower lineage scores.*

Authors’ response: We apologize for the lack of clarity in describing how the lineage score was calculated. The lineage score is intended to generate a numerical index for each cell/nucleus profile within our dataset along the continuum of progenitor to myogenic differentiation, rather than quantitate whether the cells are from a myogenic or non-myogenic lineage. We have revamped the paragraph within the Results describing the lineage score, and added additional information in the Methods section:

“To score each RMS cell (i) along a continuum of myogenic differentiation, we defined the muscle lineage score (MLS_i). We calculated the MLS_i for each cell by subtracting the progenitor (P_i) score from the differentiated (D_i) score. Unless otherwise specified, cells were scored using the new consensus progenitor, proliferative, and differentiated markers. The scores were scaled using the ScaleData Seurat’s function to center the expression values.” (pg. 26-27)

As the reviewer suggested, we have now performed a more careful analysis of MSK74711, which was generated from a *MYOD1.pL122R* mutant rhabdomyosarcoma and had a very low lineage score (i.e., cells from this sample consistently had a higher progenitor-like transcriptome). As a counterpoint, we also provide a “zoom-in” view of 29082, which was generated from a PAX3::FOXO1 sample, and had a very high lineage score (i.e., cells from this sample had a consistently higher differentiated-like transcriptome). We include UMAP plots of these two samples and additional representative two from fusion-negative and fusion-positive tumors (**Figure 3D**). We have also highlighted the meaning of low (= progenitor) vs. high lineage scores (= differentiated) in the y-axis legend.

We have added text in the Results section relevant to this analysis, and have included several sentences in the Discussion encouraging future work studying heterogeneity within *MYOD1*-mutant RMS.

“Despite overall trends in lower muscle lineage scores in FN-RMS compared with FP-RMS, we did observe considerable inter-tumoral variability of the muscle lineage scores across tumors (**Figure 3D**). For example, we identified FP-RMS with exceptionally high muscle lineage scores including 20082 and SJRHB013759_A2. By contrast, FN-RMS SJRHB010928_R1, a pre-treated FN-RMS, and MYOD1^{L122R}-mutant MSK74711 had exceptionally low muscle lineage scores.” (pg. 10)

4) It would strengthen the manuscript to clarify whether the unique presence of the neuronal-profile cell clusters in FP-RMS tumors (but not in cell lines) were the result of anatomic location and/or resection versus an intrinsic property of the FP-RMS cells. Is it possible to verify that expected RMS alterations are in the same cell populations that are classified as neuronal? Addressing these questions could reveal key mechanisms, and overcome a limitation of the study.

Authors’ response: We appreciate these suggestions, and agree that the anatomic location and treatment status are important covariates. We have included clinical data about the anatomic site where samples originated from and whether they were obtained after recent exposure to chemotherapy (as an expanded **Table S1**).

There are several lines of evidence demonstrating that neuronal cells are truly malignant:

1. These cells are transplantable and are maintained in xenografts. In our large experience with patient-derived xenografts of RMS [3-5], non-malignant stroma do not persist during xenotransplantation and passaging.

2. In the preceding studies that generated patient tumor single-cell/nucleus data (Patel et al, Dev Cell 2022 and Wei et al, Nat Cancer 2023), the authors annotated malignant or non-malignant cells using either copy-number inference or Cellassign combined with clusterprofiler enricher analysis. In our study, we have carried forward “malignant/non-malignant” annotations from the original studies to ensure that we only focus on those cells/nuclei that are from the malignant compartment of the tumor.
3. We have assessed the FP-RMS combined datasets for expression of a previously published signature of target genes for PAX3-FOXO1 [6]. We found strong expression of PAX3::FOXO1 target gene signatures across neuronal cells, suggesting that the fusion transcriptional program coincides with in this tumor subpopulation.

We have added this result into the text as a new figure panel (**Figure S4C**), and expanded the Methods and Results section to cover these findings:

“In their original studies, Wei et al. and Patel et al. both used inference of copy-number alteration analysis on the patient-derived datasets to differentiate between malignant cells (i.e. cells/nuclei harbor tumor-specific copy-number alterations) and non-malignant cells. In this study, we included only single-cell/nucleus profiles that were annotated as malignant in their respective original papers.” (Methods section; pg. 23)

*“To better characterize the FP-RMS neuronal cell state, we first performed gene set enrichment analysis with highly expressed genes found in this cell cluster and confirmed enrichment of genes associated with neurogenesis pathways including axonogenesis (GO:0007409), central nervous system development (GO:0007417) and central nervous system neuron differentiation (GO:0021953; **Table S6**). Second,*

we scored each FP-RMS subpopulation for the activity of PAX3::FOXO1 fusion oncogene using a previously defined list of fusion target genes [7]. We found that neuronal cells scored for the highest activity of fusion oncogene activity (**Figure S4C**). Last, we performed immunohistochemistry validation on PDX tumor tissue (n = 5) and confirmed in situ expression of the neuronal marker synaptophysin (SYP), which correlated with fraction of neuronal cells detected in our single-cell meta-analysis (**Figures 4B and 4C**).” (pg. 11)

5) Furthermore, it is unclear which samples were determined to have the neuronal signature. Table S7 only lists the neuronal percentage per sample, but it is not evident where the cut-off is—perhaps the samples called as containing a significant neuronal percentage should be highlighted. Furthermore, Table S6 shows that the neuronal cluster has 3/10 or 4/10 significant neural terms in the GSEA, and muscle-related terms are just as/more common. Maybe this should be noted in the text, to emphasize that these are cells with multiple significant gene expression patterns.

Authors’ response: Thank you for the suggestions. As the reviewer suggested, we have updated **Table S7** to highlight those samples that had appreciable neuronal cells, and we have updated the text to better describe our cutoff (>1% of total malignant cells in the neuronal cell state).

To address the question about GSEA analysis in **Table S6**, we have expanded our analysis of the cell states in FP-RMS. We now include a panel with a dot plot (**Figure S4B**) showing the score for each signature across the tumor cell states.

We draw the reviewer’s attention to the neuronal and differentiated cell subpopulations, which score highly for their respective signatures, implying that they are distinct

subpopulations. We also provide heatmap visualizations of differentially expressed genes (from **Table S5**) for each of the molecular subtypes as a new figure panel (**Figure S4A**), and highlight that differentiated cells express high levels of late muscle markers (e.g., *MYOG*, *TTN*, *MYH3*) while neuronal cells express high levels of neural and neural crest markers (e.g., *L1CAM*, *CHGA*, *SYP*).

The main text has been modified to report these results:

“Our combined large cohort of RMS samples enabled us to evaluate heterogeneity within FN-RMS, PAX3::FOXO1 FP-RMS and PAX7::FOXO1 FP-RMS translocated tumors as distinct entities. As expected, we identified progenitor, proliferative and differentiated cell subpopulations in each molecular subtype (**Figures 4A and S3A**). Yet, we also identified unexpected differences in gene expression between progenitor cell populations from FP-RMS and FN-RMS (**Figure S3B**) and two new subtype specific gene expression clusters (**Tables S5 and S6**). In particular, we found: (1) a group of cells/nuclei in FN-RMS that express interferon response genes such as *ISG15* and *IFIT1-3* (“IFN” cluster; 1.5% of total cells/nuclei); and, (2) a transcriptionally-distinct tumor subpopulation in FP-RMS tumors that expresses neuronal marker genes including *DCX*, *L1CAM*, *SYP*, and *CHGA* (“neuronal” cluster; 1.4% and 4.8% of total cells/nuclei from PAX3::FOXO1 and PAX7::FOXO1 FP-RMS, respectively, **Figures 4A, S4A and S4B**).” (pg. 10-11)

6) In FN-RMS, is the expression of RAS-mutants or MYOD-mutants found across clusters (e.g., progenitor, proliferative, differentiated, ground state), or are driver alleles only expressed in select populations of cells before or after therapy?

Authors' response: We agree that the intersection of genomic drivers and heterogeneity is an important question moving forward. Unfortunately, the single-cell technology used for this study, 10x Genomics' 3'-gene expression, generates sequencing data oriented at the 3' most region of the gene transcripts. Because of this, we have limited gene coverage of the 5'-most portions of most transcripts. As such, this dataset is not the appropriate one for addressing that question. We have expanded the Discussion to propose future efforts to combine analysis of the presence of genetic drivers and heterogeneity. Moreover, we suggest new and emerging techniques that may be viable options to approach this question in the future:

“Our cohort included a diversity of datasets generated from patient tumors, PDXs, primary cultures, and commercially available cell lines. Consistent with earlier reports [5, 8, 9], we note that PDXs maintain the underlying heterogeneity of patient tumors. PDXs expanded and processed at two different institutions present similar diversity of cell subpopulations (Figure S1D), indicating that PDXs represent a reproducible experimental model for studying RMS heterogeneity. In contrast, both primary and commercial cell lines were enriched for the most proliferative compartment of RMS tumors and were depleted of FP-neuronal cells (Figure S3C), suggesting that caution must be applied in using cell lines to model therapy response. The emergence of 3-dimensional culture models of RMS [1, 10, 11] may present a potential “middle ground” for in vitro models that may more faithfully recapitulate the underlying heterogeneity of RMS; recent work in DeMartino et al. indicate that organoids preserve the malignant cell states of RMS, with absence of non-malignant cells of the tumor microenvironment [1]. Genetically-engineered models represent an alternate approach to model RMS, and multiple genetic models of RMS and have been generated in mice and zebrafish [12-17]. It remains an open question of to what degree these engineered models mimic the heterogeneity of human RMS. We anticipate that the expression signatures and lineage score generated within this study will be applicable as a future tool for comparing genetically engineered models of RMS to that of patient samples.” (pg. 18)

7) *There is no indication that the code for generating the analysis is publicly available; this should be added to the manuscript. However, the methods clearly explain the analysis and should be reproducible for anyone familiar with scRNA-seq. The authors are commended for providing the extensive supplementary tables listing gene signatures for various clusters, etc to provide a resource for other labs. Please include a code repository for the analysis in the revision.*

Authors' response: We have uploaded all code from this study to a Github repository: <https://github.com/Sara-Danielli/RMS-metadata>, and uploaded R objects to FigShare: https://figshare.com/projects/RMS_consensus_analysis/194417. These links have been added to the Data and Code Availability section of the manuscript as noted below. Finally,

we have carefully edited the Methods section to describe and cite all computation tools and packages used in this study.

"The code to reproduce the main results included in the paper is available at <https://github.com/Sara-Danielli/RMS-metadata>. The RMS single-cell objects generated in this study have been uploaded on FigShare: https://figshare.com/projects/RMS_consensus_analysis/194417" (pg. 31)

Minor Comments

1) Reference #21 is a Biorxiv pre-print and should be updated to reflect the most recent form of that work.

Authors' response: Apologies for the error. This has been corrected in the revised manuscript.

2) A more in depth characterization of "transiting-differentiated" cells would strengthen the manuscript. What does "had lower levels of ... cells states genes" mean? Should also correct the typo here.

Authors' response: We have added an additional panel (**Figure S2A**) with violin plots showing that the transiting differentiated and transiting progenitor populations have intermediate scores for the differentiated and progenitor signatures, respectively. We have corrected the typographical error in the revised manuscript, and we have modified the main text to more clearly reflect this finding:

*"By comparing the progenitor and differentiated signature scores to our categorical cell subpopulations, we found that both the progenitor and the differentiated signatures showed a gradient of expression across the molecularly defined subpopulations (**Figure S2A**). Moreover, most cycling cells/nuclei preferentially mapped to cells with low progenitor and differentiated scores, irrespective of the RMS subtype (**Figure S2B**)."* (pg. 9)

3) The section heading “The muscle lineage score stratifies between RMS subtypes” anthropomorphizes and should be revised. Perhaps an alternative like “The muscle lineage score reveals key distinctions between RMS subtypes” would be more clear.

Authors’ response: We have changed the section header to “Muscle lineage score reveals key distinctions between RMS subtypes”, as the reviewer suggested (pg. 9).

4) The statement in lines [219-221] “Yet, when we analyzed the overlap in expressed genes from these seemingly shared states, we observed that progenitor cell states were not transcriptionally the same across each tumor subtype (Figure S3B)...” is unclear as written. Does this mean “across” subtypes as in comparing ERMS to ARMS, or across samples within a genetically defined subtype? Revision to clarify would strengthen them manuscript.

Authors’ response: We have rephrased this section for clarity:

“...we identified progenitor, proliferative and differentiated cell subpopulations in each molecular subtype (**Figures 4A and S3A**). Yet, we also identified unexpected differences in gene expression between progenitor cell populations from FP-RMS and FN-RMS (**Figure S3B**) and two new subtype specific gene expression clusters (**Tables S5 and S6**).” (pg. 10-11)

5) On line 446, a reference needs to be filled in.

Authors’ response: This has been corrected in the revised manuscript, now located on page 23.

6) The model figure, Figure 6, is very clear and communicates the main points of the manuscript well. However, the differentiated population, which has an increasing signature early with therapy in FP-RMS according to Figure 5, is not indicated in this population in FP-RMS. Only the neuronal and FP-progenitors are shown. This model figure should be updated so it fully reflects the conclusions of the manuscript.

Authors' response: Figure 6 has been modified, as the reviewer suggested, to more accurately reflect the results in Figure 5.

7) In lines 311-312, the authors state “these studies identify important new cell states that are retained and expanded after therapy.” It is important to note that bulk RNA-seq is used in Figure 5, and so the data does not necessarily indicate the proportion of cells of each type. This should be clarified in the text to aid the reader, given that Figure 6 assumes the proportion of each population has been determined on-therapy and at relapse.

Authors' response: In reponse to this comment, we have now modified Figures 5 and S6, with additional analysis on our single-cell RNA datasets of 4 untreated, and 7 treated FN-RMS samples, showing the same trend that Progenitor in FN-RMS is enriched, Differentiated score is lower in treated samples (Figure S6A). In addition, in one matched sample SJRHB000026, we showed that the same trend was observed, with Progenitor score significantly upregulated in relapse tumor (Figure 5A). Yet, the patient samples from diagnostic biopsy are challenging to collect fresh to run single-cell RNA-sequencing. Therefore, we only have 7 matched FN-RMS, and 2 matched FP-RMS with bulk RNA-seq results with the application of our signatures (Figure 5C).

REVIEWER #2:

The team provides a nice view of RMS expression by developing a single cell atlas across primary tumors and cell lines. The dataset is a tour de force especially because there are were many unique patient samples. They define four populations: progenitors, proliferative, differentiated, and ground cells. They stratify these RMS cells along the continuum of human muscle development and show that some RMS cells share expression patterns with fetal/embryonal myogenic precursors rather than postnatal satellite cells. They also identify populations resistant after chemotherapy and new states in the RMS cells not in development. Overall, an important and interesting study for the field. Few comments/questions below:

1. FN-RMS closely resemble bipotent SkM.Mesenchymal cells-how? Based on both marker expression and function? How consistent/what are the differences in these two populations? Can the patient derived SkM.Mesenchymal cells differentiate similar to fetal developmental cells or different?

Authors' response: We thank the reviewer for these questions. We have used the skeletal mesenchymal (SkM.Mesenchymal) gene expression signature from Xi et al. [18] and applied it to our combined FN-RMS dataset. We show in **Figure 4E** that the majority of FN-Progenitor cells map to Skeletal mesenchymal cells using SingleR, an unbiased cell type annotation approach that generates assignment scores using the Spearman correlation between query (RMS single-cell data) and reference (Xi et al. muscle development atlas) cells/nuclei.

Moreover, we now show in **Figure S5C** that the FN-RMS progenitor cells score highly for marker genes of the SkM.Mesenchymal population (*OGN*, *THY1*, *POSTN*).

These results have been included in the Results section:

*“In particular, FN-Progenitor cells preferentially mapped to SkM.Mesen cells and expressed high levels of marker genes of this developmental subpopulation (*OGN*, *THY1*, *POSTN*) (Figures 4E and S5C). (pg. 13)*

Moreover, we have expanded the Discussion to encompass functional similarities between the FN-RMS progenitor cells and SkM.Mesenchymal cells, namely their ability to differentiate into osteogenic lineages when cultured in bone conditioning media [8] and their ability to re-populate the myogenic hierarchy after flow sorting [5, 8].

“Intriguingly, ‘Progenitor’ FN-RMS cells shared gene expression similarity to a newly defined Skeletal muscle mesenchymal cell state that has bipotent capability to make muscle and osteogenic lineage cells [18]. Based on functional studies showing that this RMS cell state can drive tumor growth after stress and has the capacity to make osteogenic lineage cells [8], we have refined our naming of this cell state as “FN-SkM.Mesen-like” (pg. 19)

2) What percentage of SkM.Mesenchymal cells are found in FN patient samples compared to the percentage seen in human development? How varied is this across different patient samples? In the supplement can you show the % of the different cell types present in the FN samples and variability or similarities?

Authors’ response: As the reviewer suggested, we provide a deeper analysis of the SkM.mesenchymal cells within both fetal muscle and within our RMS cohort. We have added two figure panels to the manuscript and edited the text:

“Of note, the number of SkM.Mesen cells peak at 12-14 weeks of development, where they comprise 23.5% of the fetal myogenic cells (**Figure S5B**). Similarly, we measured 15.7% of cells/nuclei within FN-RMS mapped to SkM.Mesen cells (**Figure 4E and Table S9**). This contrasts with FP-RMS that largely lack cell state similarity with SkM.Mesen cells (1.4% and 0.3% of PAX3::FOXO1 and PAX7::FOXO1 samples, respectively). These results are also in keeping with the identification of differences in gene expression between progenitor cell populations from FP-RMS and FN-RMS (**Figure S3B**).” (pg. 13)

All of this information is also incorporated into **Table S7**. The above panel, which is now **Figure S5B**, shows the percentage of different skeletal muscle cell types across human development (Xi et al.; [18]). Moreover, we have added panel **Figure 4D**, which shows sample-by-sample variation of cell proportion that map to each myogenic developmental

state across RMS samples. We find that this panel visualizes the striking difference in developmental patterning between FN-RMS and FP-RMS.

3) Are there candidate targets that could provide novel molecular markers for less differentiated progenitor cells compared to differentiated muscle-like cells? Are the current therapies targeting either of these populations? Or does this dataset provide new molecular targets for treatment? I.e. could be discussed

Authors' response: Yes, we wholeheartedly agree with the reviewer. A strength of our analysis is the introduction of potential molecular markers, especially candidate cell surface markers, for both addressing research questions and for therapeutic targeting. We have cross-referenced the differential gene expression markers in our consensus analysis (**Table S8**) with known cell surface proteins from the Human Protein Atlas and highlighted them in their respective tables. We have added text to the revised manuscript to highlight this:

“Lastly, we sought to identify potential candidate cell surface markers in our consensus analysis that could be used to both address future research questions and/or therapeutic targeting. We cross-referenced the gene expression markers identified across each tumor subpopulation for each RMS entity with known cell surface proteins from the Human Protein Atlas. We identified several cell surface markers, including CD44 for Progenitor, ERBB3 for Differentiated, and L1CAM for Neuronal cells (Table S8).” (pg. 12)

4) There is a bipotent NMP in development that gives rise to both muscle and neuronal. Does the FP RMS or FN population/s express any of these markers at any time point?

Authors' response: We looked for expression of brachyury (*TBXT*) and *SOX2* within our single-cell dataset. We were unable to detect any appreciable expression in the combined (all RMS) or separated FP-RMS dataset, and provide those results here.

We have chosen to restrict this result to the reviewer reponse, as it does not fit well with the flow of this manuscript.

5) Do they (neural cell states) revert back to this after treatment? Or thoughts on how this neural population arises/what it is?

Authors' response: We agree that these questions, provoked by the findings within this study, will be productive areas for future investigation. In particular, we agree that the question of lineage transition is a very important question for in-depth study. Understanding the mechanics of myogenic-to-neuronal lineage will, undoubtedly, improve our understanding of FP-RMS and has the potential to inform future therapeutic directions. Yet, these studies are currently beyond the scope of this manuscript.

We have expanded on the discussion and highlighted what we learned in this study, the current state of the literature, and recommended future directions. Moreover, we draw threads to other cancers, namely adult cancers (e.g., castration-resistant neuroendocrine prostate cancer, colorectal cancer, lung cancer) where transdifferentiation to a neuroendocrine phenotype has emerged as a mechanism of treatment escape.

“Our analysis also uncovered that FP-RMS do not display the same rigid developmental hierarchies as found in normal development and may contain different therapy persister cell states. For example, FP-RMS tumors have fewer overall proportions of progenitor cells with some tumors seemingly lacking this cell state completely. Moreover, although the FP-progenitor cells do express mesenchymal markers, they are transcriptionally distinct from the SkM.Mesen cells found in fetal muscle development and the FN-SkM.Mes-like state discovered here. Indeed, DeMartino et al., also independently identified key differences in mesenchymal-pathway enriched cell states in comparing scRNA sequencing expression of FP- and FN-RMS [1]. In addition, a subset of FP-RMS have tumor cells that have neuronal-pathway activation that are not found in human muscle development and yet are enriched after chemotherapy. The existence of this FP-RMS cell state is supported by immunohistochemical studies of 42 FP-RMS tumors that identified a subset of FP-RMS express marker genes including chromogranin, CD56, and synaptophysin [19]. While RMS have demonstrated histological resemblance to cells of the myogenic lineage [20-22], our work suggests that FP-RMS are able to transition to cell states not found in normal myogenic development. Intriguingly, lineage plasticity and neuroendocrine transdifferentiation have been reported as resistance mechanisms in multiple adult cancer types, including melanoma and castration-resistant prostate cancer [23-25]. Indeed, earlier experiments using limiting dilution cell transplantation assays into immune deficient mice showed that FP-RMS have a high frequency of tumor initiation, raising the possibility that most if not all FP-RMS cells can acquire the ability to propagate tumors in vivo [26]. The frequency by which this tumor-acquired cell states are found in FP-RMS and defining their possible role in driving therapy resistance will clearly be a major research focus for the field in the future.” (pg. 20-21)

6) What about overlap of the RMS neuronal datasets with neural crest or neuroendocrine cells? I.e. Further evaluate what neural cells are these similar to?

Authors' response: Thank you for bringing up this point. In fact, we do note expression markers of neural crest development within the 'RMS neuronal cells' including expression of classic markers of sympathoadrenal development (e.g., synaptophysin (*SYP*), chromogranin A (*CHGA*), *PHOX2A* (*PHOX2A*); **Figure S4A** and **Table S5**). In fact, we now show that the neuronal subpopulation can be detected using IHC staining of synaptophysin, which is positively correlated with fraction of neuronal cell fraction by single-cell meta-analysis (**Figures 4B** and **4C**).

We have highlighted these genes in new heatmaps (**Figure S4A**, see below). Moreover, we analyzed the signature scores including Neuronal signature across distinct subpopulations and found that Neuronal signature is enriched only in Neuronal subpopulation but not others (**Figure S4B**, see below), suggesting it is a distinct subpopulation.

7) Would be interesting to see if the entire PAX3 or PAX7 networks are activated in the FP-RMS cells with downstream single cell analysis platform or chip-seq or cut and run/tag etc.

Authors' response: We have used existing datasets of target genes for the PAX3-FOXO1 fusion oncogene. Each FP-RMS cell/nucleus was scored for activity of each transcription factor. The results are shown below:

*“Second, we scored each FP-RMS subpopulation for the activity of PAX3::FOXO1 fusion oncogene using a previously defined list of fusion target genes [7]. We found that neuronal cells scored for the highest activity of fusion oncogene activity (**Figure S4C**).”* (pg. 11)

8) Is there a way to target cells within the continuum of cancer states identified from this dataset? If so perhaps a discussion of which of these states may be best to target should be included?

Authors' response: Yes, we agree this is a key discussion point moving forward and have expanded our text:

*“Lastly, we sought to identify potential candidate cell surface markers in our consensus analysis that could be used to both address future research questions and/or therapeutic targeting. We cross-referenced the gene expression markers identified across each tumor subpopulation for each RMS entity with known cell surface proteins from the Human Protein Atlas. We identified several cell surface markers, including CD44 for Progenitor, ERBB3 for Differentiated, and L1CAM for Neuronal cells (**Table S8**).”* (pg. 12)

9) *What overlap do the mouse tumor models have with the different populations identified? ie are they good models?*

Authors' response: Yes, we agree this is an important future direction. Unfortunately, very few genetically-engineered mouse models of RMS have undergone single-cell sequencing, and generating those datasets are beyond the scope of our present study. We believe that our unified dataset provides a framework of 'ground truth' data for future study in that direction. We have expanded the Discussion to broadly cover what we have learned in this study by comparing patient samples, PDXs, and cell lines. Moreover, we have included text about emerging models such as 3D culture models, and we lay out a rationale for future comparison of our dataset to genetically engineered models of RMS:

“Our cohort included a diversity of datasets generated from patient tumors, PDXs, primary cultures, and commercially available cell lines. Consistent with earlier reports [5, 8, 9], we note that PDXs maintain the underlying heterogeneity of patient tumors. PDXs expanded and processed at two different institutions present similar diversity of cell subpopulations (Figure S1D), indicating that PDXs represent a reproducible experimental model for studying RMS heterogeneity. In contrast, both primary and commercial cell lines were enriched for the most proliferative compartment of RMS tumors and were depleted of FP-neuronal cells (Figure S3C), suggesting that caution must be applied in using cell lines to model therapy response. The emergence of 3-dimensional culture models of RMS [1, 10, 11] may present a potential “middle ground” for in vitro models that may more faithfully recapitulate the underlying heterogeneity of RMS; recent work in DeMartino et al. indicate that organoids preserve the malignant cell states of RMS, with absence of non-malignant cells of the tumor microenvironment [1]. Genetically-engineered models represent an alternate approach to model RMS, and multiple genetic models of RMS and have been generated in mice and zebrafish [12-17]. It remains an open question of to what degree these engineered models mimic the heterogeneity of human RMS. We anticipate that the expression signatures and lineage score generated within this study will be applicable as a future tool for comparing genetically engineered models of RMS to that of patient samples.” (pg. 18)

10) *Is the fact that you see an increase in treatment-induced selection for the FN-SkM.Mes-like progenitor state in FN- RMS a good thing or do these cells then lead to cancer relapse? Do you see an increase in FN-SkM.Mes-like cells after treatment in FN-patients +/- treatment or in multiple relapses? Can this be quantified?*

Authors' response: This is an important point. As shown in **Figures 5D and S6**, we use matched patient tumors to show that treatment have significantly increased FN.progenitor signature scores when treated. Moreover, xenografts and cell lines generated from these exact patient samples have significantly higher (albeit less dramatic) FN.progenitor scores, implicating that FN-RMS cells acquire a stable progenitor-rich phenotype after therapy.

Finally, we have expanded the discussion to cover treatment-induced persisters, and work from both Patel et al and Wei et al, indicating that the FN-SkM.MES-like cells expressing CD44, CD90, and/or EGFR were enriched during therapy and capable of repopulating xenografts after flow-sorting. Indeed, in a prior study, treated FN-RMS tumors were enriched for MEOX2, which demarcates the FN-RMS progenitor cell state (Patel et al, *Dev Cell* 2022).

“Our findings also contribute to a growing body of literature describing rare cancer cells with the capacity to propagate and re-establish tumors after therapy, which have sometimes been called cancer stem cells or tumor-propagating cells [27-29]. We identified a group of cells expressing a progenitor signature including markers of early muscle progenitors such as CD44, EGFR, and THY1 (CD90). Previous work using flow sorting for cell surface markers such as CD133, CD44 and EGFR have validated the existence of these cells in FN-RMS and demonstrated that they propagate FN-RMS both in vitro and when grown in immunocompromised mice [8, 30-35]. Intriguingly, ‘Progenitor’ FN-RMS cells shared gene expression similarity to a newly defined Skeletal muscle mesenchymal cell state that has bipotent capability to make muscle and osteogenic lineage cells [18]. Based on functional studies showing that this RMS cell state can drive

tumor growth after stress and has the capacity to make osteogenic lineage cells [8], we have refined our naming of this cell state as “FN-SkM.Mesen-like”. Our analysis also suggested that FN-RMS replicate the broad diversity of fetal muscle development cell states and have a shared developmental hierarchy with early developing fetal muscle found after 7 weeks post-conception. These findings contrast with Patel et al. which proposed that FN-RMS recapitulate an earlier mesodermal specification program in mice [30]. This difference is likely attributable to interspecies variation in myogenesis, especially since bipotent SkM.Mesen cells have yet to be identified in mice, or that comparison with human muscle development did not include cell types from the earliest stages of mesodermal specification that begin at 24 days post-conception in humans. Finally, our data suggests that FN-SkM.Mesen-like cells are largely quiescent and are likely the therapy persistent cells that re-establish tumors after treatment. Indeed, a similar phenomenon where cells with characteristics of progenitors from the hematopoietic, colon, and brain lineages have been proposed to play roles in leukemia, colorectal cancer, and glioblastoma, respectively [36-38].” (pg. 18-19)

11) Discussion- could any of these new populations be used as biomarkers for potential to relapse? Or that therapy treatment did/did not work well?

Authors’ response: We agree that this novel markers of treatment resistance would be powerful tools for RMA therapy. We have attempted to perform biomarker analysis by applying our signatures to existing microarray data from the ITCC consortium [39]. To our knowledge, this is the only RMS dataset that provides RNA expression level data alongside outcome data. We provide results for the reviewer response for FN-RMS and FP-RMS on the next page, using the mean signature score as a cutoff.

In a nutshell, the data is underpowered to make definitive statements about using signatures for prognosis. Overall, we think this underscores the need for larger high-quality RMS datasets combining molecular and survival information (such as datasets being currently generated by the CCDI and the COG EveryChild initiatives). Because the outcome data above is difficult to interpret due to power limitations, we have restricted these figures to the reviewer response.

REVIEWER #3:

This is a well written and constructed study of rhabdomyosarcoma single cell tumor data. The authors describe a unified analysis of each groups previously published separately generated and analyzed data. They report an interesting spectrum of gene expression patterns consistent with muscle lineage; therapy resistant progenitor states in fusion negative rhabdomyosarcoma and an interesting neuronal subtype associated with the fusion positive histology. This will be an important reference work for the field, so I strongly support the effort. However, I identified a few things that may be useful in improving the study.

1. The first step that they authors take is to generate an integrated atlas of all rhabdomyosarcoma cells. This seems like an odd first step after spending much of the introduction telling us that their interest is defining differences. As the authors know the anchor based integration may be forcing cells into groups. What happens if you overlaid cells from another embryonal histology? Would the Louvian method force those cells onto the same atlas?

Authors' response: We appreciate the reviewer's comment about integration analysis. We began this effort with the goal of unifying analyses from three independent studies from Patel et al., Danielli et al., and Wei et al, who individually reported different patient samples with different bioinformatics tools, but identified similar subpopulations with confusing nomenclatures. Therefore we chose, among a number of integration strategies, to perform Seurat's anchor-based RPCA method because it is less prone to over-correction [40]. However, all integration schema run the risk of removing a degree of biological variation in favor of batch correction. The challenge of over-correction is, in part, the inspiration for **Figure 3**. **Figure 3** is entirely generated on a sample-by-sample basis and shows that the signatures and lineage score perform robustly independent of integration or batch-correction.

We also provided the rationale for our choice of Louvain clustering resolution below in our response to comment #3.

2. What happens if you separate the fusion positive and fusion negative and then perform the Louvian clustering? Do you get the same answer?

Authors' response: We apologize for the confusion. The analysis reported in **Figure 4** were performed using the method the reviewer proposed. Briefly, we separated molecular groups (fusion-negative, PAX3::FOXO1, and PAX7::FOXO1) and integrated each group

separately. Louvain clustering was used to annotate the separate subpopulations (**Table S5**). We have modified the manuscript to more clearly reflect our approach:

*“Our initial analyses centered on combining RMS subtypes together to identify conserved cell states shared across pediatric RMS. While this approach enabled us to define key muscle-lineage cell states shared across RMS, it would likely fail to identify subtype-specific subpopulations or differences in gene expression within defined subpopulations in RMS subtypes. Our combined large cohort of RMS samples enabled us to evaluate heterogeneity within FN-RMS, PAX3::FOXO1 FP-RMS and PAX7::FOXO1 FP-RMS translocated tumors as distinct entities. As expected, we identified progenitor, proliferative and differentiated cell subpopulations in each molecular subtype (**Figures 4A and S3A**).”*
(pg. 10)

3. I would like to see the clustering based on the number of genes and justification for the resolution used to create the Louvian clusters. 11 clusters may be too high of a resolution for what they are seeing.

Authors' response: As the reviewer suggested, we performed Louvain clustering in the shared neighborhood network at a variety of resolutions. As you can see, the number of identified clusters stabilizes at a resolution of around 0.3, yielding a total of 12 clusters. For this reason, we performed follow-up analyses using a resolution of 0.3.

We have added text in the revised manuscript Methods part to clarify the inspiration and reasoning for this approach:

“...We built a K-nearest neighbor (KNN) graph, used the Louvain algorithm for clustering the cells (resolution of 0.2-0.3), and visualized the cells using UMAP plots. The number of identified clusters stabilizes at a resolution of around 0.3, yielding a total of 12 clusters. For this reason, we performed our analyses using a resolution of 0.3...” (pg. 25)

4. The conclusion then that there are differences between the TR-differentiated and differentiated may be overly-fit. If they are making the point that they see a difference here it would be nice to see if these groups were biologically real.

Authors' response: Indeed, we were surprised to see that the gene expression patterns between the differentiated and TR-differentiated represented a gradient, suggesting that RMS cells undergo a process of transition across a broad spectrum of myogenic states. Ultimately, determining the degree of resolution by which to perform “hard” clustering methods such as Louvain clustering represent a qualitative decision. As a result, we developed the lineage scoring metric in **Figure 3** to obviate this challenge and to serve as an integration- and clustering-independent method.

5. *Figure 3 shows a muscle lineage score continuum. I like the concept but think the definition of AU Differentiated – AU Progenitor score is too crude and may be misleading. At a minimum the gene programs that are dynamically driving this observation should be shown in Fig 3.*

Authors' response: We have expanded both the main text and the methods section to better explain the lineage score. Moreover, we have provided more information in the main text to explain the reason for developing the lineage score, namely that it is a continuous readout that avoids having to place arbitrary cutoffs, as the reviewer pointed out above in comment #1.

*“...By comparing the progenitor and differentiated signature scores to our categorical cell subpopulations, we found that both the progenitor and the differentiated signatures showed a gradient of expression across the molecularly defined subpopulations (**Figure S2A**). Moreover, most cycling cells/nuclei preferentially mapped to cells with low progenitor and differentiated scores, irrespective of the RMS subtype (**Figure S2B**). These results led us to create a “muscle lineage score,” defined as the difference between the differentiated and progenitor signature scores, and to apply this scoring metric to every single-cell/nucleus profile within our atlas in relation to their proliferation properties....” (pg. 9)*

Moreover, we provide a heatmap for the reviewer showing that the lineage score stratifies RMS samples and cells:

6. One idea is that the analysis in Figure 4 is a much more accurate representation of the heterogeneity of the dataset. Again, I worry that Fig 3 has collapsed the data to a lowest common denominator which may or may not reflect reality. I might suggest moving Fig3 to the supplement.

Authors' response: The strength of the lineage score in **Figure 3** is that it was performed entirely independent of a priori knowledge (e.g., fusion status, whether the tumor was obtained after treatment). As such, we think there's value in **Figure 3** as a "blind" classification method. In contrast, **Figure 4** represents our effort to better understand the differences in heterogeneity across molecular subtypes of RMS. As such, we have chosen to maintain the narrative structure of the manuscript. We have modified the text to better support the narrative flow:

"Our initial analyses centered on combining RMS subtypes together to identify conserved cell states shared across pediatric RMS. While this approach enabled us to define key muscle-lineage cell states shared across RMS, it would likely fail to identify subtype-specific subpopulations or differences in gene expression within defined subpopulations in RMS subtypes. Our combined large cohort of RMS samples enabled us to evaluate heterogeneity within FN-RMS, PAX3::FOXO1 FP-RMS and PAX7::FOXO1 FP-RMS translocated tumors as distinct entities.." (pg. 10)

In addition, we have provided additional validation analysis on a new scRNA-seq dataset from DeMartino research [1], 19 patient-derived samples with no overlap with our datasets, with plate-based SORT-seq technologies (**Figure S2D**), validating the value of the muscle lineage score, where it can distinguish between FN-RMS and FP-RMS:

7. They mention several times that the Patel Myoblast cells were presumably absorbed by another cluster in the current analysis. Where did they go? Are they not real?

Authors' response: We appreciate the reviewer's comment. To be brief, reports from three independent research employed distinct methods with limited numbers of patient samples to annotate subpopulations, with a potential overcall of a signature (myoblast), or an undercall which led to too many subpopulations that are not shared across different patients. In the case of the Patel myoblast population, the defined signature from the work failed to identify a distinct cluster of cells within the combined datasets. We conclude that the "myoblast" signature from the Patel paper does not define a specific cell state. This discrepancy underscores the value of a unified study, such as this one, for the purpose of resolving differences in analytical approaches.

8. Given the implications of the described interferon response cells in FN-RMS and the rare neuronal cell population in FP-RMS I would ask the authors for validation using correlative assay such as RNAish or if possible IHC.

Authors' response: Thank you for requesting this important set of additional, clarifying experiments. We have now used synaptophysin, which is expressed in FP-RMS neuronal cells as an IHC stain, and performed IHC of patient-derived xenografts. We show an example of IHC demonstrating that FP-RMS xenografts have synaptophysin positivity, and that staining in FP-RMS linearly correlates with the single-cell RNA-seq data.

These panels have been added to **Figures 4B and 4C**, and we have added text in the revised manuscript:

*“To better characterize the FP-RMS neuronal cell state, we first performed gene set enrichment analysis with highly expressed genes found in this cell cluster and confirmed enrichment of genes associated with neurogenesis pathways including axonogenesis (GO:0007409), central nervous system development (GO:0007417) and central nervous system neuron differentiation (GO:0021953; **Table S6**). Second, we scored each FP-RMS subpopulation for the activity of PAX3::FOXO1 fusion oncogene using a previously defined list of fusion target genes [7]. We found that neuronal cells scored for the highest activity of fusion oncogene activity (**Figure S4C**). Last, we performed immunohistochemistry validation on PDX tumor tissue (n = 5) and confirmed in situ expression of the neuronal marker synaptophysin (SYP), which correlated with fraction of neuronal cells detected in our single-cell meta-analysis (**Figures 4B and 4C**). Overall, these results confirm that FP-RMS tumors contain a unique subpopulation of tumor cells that expresses markers of neuronal cells.”* (pg. 11)

9. The therapy resistant cells may be the most impactful finding of the study. I struggle again with why the authors collapse the data and figure into FN and FP and metaprograms. Why not have a figure that broadens the analysis of the FN patient samples and a separate one using the PDX model?

Author’s comments: Indeed, we agree that the therapy-resistant portion of the study has the potential to be clinically impactful. As the reviewer suggested, we have restructured this portion of the paper and expanded **Figures 5 and S6** to separate FN- and FP-RMS samples. In a new panel, we show that FN-RMS patient samples obtained in the midst of therapy scored higher for the progenitor score compared to samples obtained before therapy (**Figure S6A**).

A
Because this analysis may be confounded by inter-patient variability, we investigate one matched pair of samples (SJRHB000026_R2 and SJRHB000026_R3) obtained from the same patient before (SJRHB000026_R2) and in the midst of therapy (SJRHB000026_R3). In a new panel (**Figure 5A**), we highlight this matched pair of samples and show that the treated sample (SJRHB000026_R3) has a dramatically higher progenitor score compared to the untreated one (SJRHB000026_R2).

A
As the reviewer suggested, we have adjusted our analysis of matched FFPE RNA-seq samples to focus on FN-RMS alone (**Figures 5C and 5D**). Consistent with our analysis in **Figures 5A**, we show that treatment increased the progenitor score in 5/7 FN-RMS patients (**Figure 5D**).

D
10. Does the PDX study need addition replicates to be significant? I think most people will agree with decrease in the proliferative gene set but that would not be entirely surprising. The novel finding would be the other gene programs. Would doing this experiment with a single cell assay readout not be a more conclusive story? Bulk RNA felt kind of disappointing.

Author's comments: As the reviewer suggested above, we have restructured the treatment-persistence portion of the manuscript to separate the FN- and FP-RMS portions. In a new supplemental panel, we show analogous data to **Figures 5D** for two matched pairs from FP-RMS patients (**Figure S6C**).

C
Due to low numbers of matched samples, the analysis is underpowered to perform statistical testing of significance. We have attempted to identify additional matched clinical specimens from FP-RMS patients, but were unsuccessful. As there are ~80 new cases

of FP-RMS in the US annually and because delayed resection is only performed for a subset of those cases, it is unlikely that we will collect additional samples within a reasonable timeframe. As such, we include language in the Discussion to summarize our observations in clinical samples and highlight the pressing need for collecting longitudinal samples from FP-RMS patients to clarify treatment persistence in this patient population.

References:

1. DeMartino, J., et al., *Single-cell transcriptomics reveals immune suppression and cell states predictive of patient outcomes in rhabdomyosarcoma*. Nat Commun, 2023. **14**(1): p. 3074.
2. Buttner, M., et al., *scCODA is a Bayesian model for compositional single-cell data analysis*. Nat Commun, 2021. **12**(1): p. 6876.
3. Stewart, E., et al., *Orthotopic patient-derived xenografts of paediatric solid tumours*. Nature, 2017. **549**(7670): p. 96-100.
4. Stewart, E., et al., *Identification of Therapeutic Targets in Rhabdomyosarcoma through Integrated Genomic, Epigenomic, and Proteomic Analyses*. Cancer Cell, 2018. **34**(3): p. 411-426 e19.
5. Patel, A.G., et al., *The myogenesis program drives clonal selection and drug resistance in rhabdomyosarcoma*. Dev Cell, 2022.
6. Gryder, B.E., et al., *PAX3-FOXO1 Establishes Myogenic Super Enhancers and Confers BET Bromodomain Vulnerability*. Cancer Discov, 2017. **7**(8): p. 884-899.
7. Gryder, B.E., et al., *Miswired Enhancer Logic Drives a Cancer of the Muscle Lineage*. iScience, 2020. **23**(5): p. 101103.
8. Wei, Y., et al., *Single-cell analysis and functional characterization uncover the stem cell hierarchies and developmental origins of rhabdomyosarcoma*. Nat Cancer, 2022. **3**(8): p. 961-975.
9. Danielli, S.G., et al., *Single-cell profiling of alveolar rhabdomyosarcoma reveals RAS pathway inhibitors as cell-fate hijackers with therapeutic relevance*. Sci Adv, 2023. **9**(6): p. eade9238.
10. Savary, C., et al., *Fusion-negative rhabdomyosarcoma 3D organoids to predict effective drug combinations: A proof-of-concept on cell death inducers*. Cell Rep Med, 2023. **4**(12): p. 101339.
11. Meister, M.T., et al., *Mesenchymal tumor organoid models recapitulate rhabdomyosarcoma subtypes*. EMBO Mol Med, 2022. **14**(10): p. e16001.
12. Searcy, M.B., et al., *PAX3-FOXO1 dictates myogenic reprogramming and rhabdomyosarcoma identity in endothelial progenitors*. Nat Commun, 2023. **14**(1): p. 7291.
13. Drummond, C.J., et al., *Hedgehog Pathway Drives Fusion-Negative Rhabdomyosarcoma Initiated From Non-myogenic Endothelial Progenitors*. Cancer Cell, 2018. **33**(1): p. 108-124 e5.
14. Nakahata, K., et al., *K-Ras and p53 mouse model with molecular characteristics of human rhabdomyosarcoma and translational applications*. Dis Model Mech, 2022. **15**(2).
15. Nishijo, K., et al., *Credentialing a preclinical mouse model of alveolar rhabdomyosarcoma*. Cancer Res, 2009. **69**(7): p. 2902-11.
16. Kendall, G.C., et al., *PAX3-FOXO1 transgenic zebrafish models identify HES3 as a mediator of rhabdomyosarcoma tumorigenesis*. Elife, 2018. **7**.
17. Yan, C., et al., *Visualizing Engrafted Human Cancer and Therapy Responses in Immunodeficient Zebrafish*. Cell, 2019. **177**(7): p. 1903-1914.e14.

18. Xi, H., et al., *A Human Skeletal Muscle Atlas Identifies the Trajectories of Stem and Progenitor Cells across Development and from Human Pluripotent Stem Cells*. *Cell Stem Cell*, 2020. **27**(1): p. 181-185.
19. Bahrami, A., et al., *Aberrant expression of epithelial and neuroendocrine markers in alveolar rhabdomyosarcoma: a potentially serious diagnostic pitfall*. *Mod Pathol*, 2008. **21**(7): p. 795-806.
20. Chen, X., et al., *Targeting oxidative stress in embryonal rhabdomyosarcoma*. *Cancer Cell*, 2013. **24**(6): p. 710-24.
21. Kahn, H.J., et al., *Immunohistochemical and electron microscopic assessment of childhood rhabdomyosarcoma. Increased frequency of diagnosis over routine histologic methods*. *Cancer*, 1983. **51**(10): p. 1897-903.
22. Skapek, S.X., et al., *Rhabdomyosarcoma*. *Nat Rev Dis Primers*, 2019. **5**(1): p. 1.
23. Zou, M., et al., *Transdifferentiation as a Mechanism of Treatment Resistance in a Mouse Model of Castration-Resistant Prostate Cancer*. *Cancer Discov*, 2017. **7**(7): p. 736-749.
24. Rambow, F., et al., *Toward Minimal Residual Disease-Directed Therapy in Melanoma*. *Cell*, 2018. **174**(4): p. 843-855 e19.
25. Davies, A., et al., *The Transcriptional and Epigenetic Landscape of Cancer Cell Lineage Plasticity*. *Cancer Discov*, 2023. **13**(8): p. 1771-1788.
26. Generali, M., et al., *High Frequency of Tumor Propagating Cells in Fusion-Positive Rhabdomyosarcoma*. *Genes (Basel)*, 2021. **12**(9).
27. Genadry, K.C., et al., *Soft Tissue Sarcoma Cancer Stem Cells: An Overview*. *Front Oncol*, 2018. **8**: p. 475.
28. Dela Cruz, F.S., *Cancer stem cells in pediatric sarcomas*. *Front Oncol*, 2013. **3**: p. 168.
29. Hettmer, S. and A.J. Wagers, *Muscling in: Uncovering the origins of rhabdomyosarcoma*. *Nat Med*, 2010. **16**(2): p. 171-3.
30. Patel, A.G., et al., *The Myogenesis Program Drives Clonal Selection and Drug Resistance in Rhabdomyosarcoma*. *bioRxiv*, 2021: p. 2021.06.16.448386.
31. Walter, D., et al., *CD133 positive embryonal rhabdomyosarcoma stem-like cell population is enriched in rhabdospheres*. *PLoS One*, 2011. **6**(5): p. e19506.
32. Radzikowska, J., et al., *Cancer Stem Cell Markers in Rhabdomyosarcoma in Children*. *Diagnostics (Basel)*, 2022. **12**(8).
33. Linardic, C.M., et al., *Genetic modeling of human rhabdomyosarcoma*. *Cancer Res*, 2005. **65**(11): p. 4490-5.
34. Ignatius, M.S., et al., *In vivo imaging of tumor-propagating cells, regional tumor heterogeneity, and dynamic cell movements in embryonal rhabdomyosarcoma*. *Cancer Cell*, 2012. **21**(5): p. 680-693.
35. Blum, J.M., et al., *Distinct and overlapping sarcoma subtypes initiated from muscle stem and progenitor cells*. *Cell Rep*, 2013. **5**(4): p. 933-40.
36. Singh, S.K., et al., *Identification of human brain tumour initiating cells*. *Nature*, 2004. **432**(7015): p. 396-401.
37. Ricci-Vitiani, L., et al., *Identification and expansion of human colon-cancer-initiating cells*. *Nature*, 2007. **445**(7123): p. 111-5.
38. Lapidot, T., et al., *A cell initiating human acute myeloid leukaemia after transplantation into SCID mice*. *Nature*, 1994. **367**(6464): p. 645-8.

39. Davicioni, E., et al., *Identification of a PAX-FKHR gene expression signature that defines molecular classes and determines the prognosis of alveolar rhabdomyosarcomas*. *Cancer Res*, 2006. **66**(14): p. 6936-46.
40. Hao, Y., et al., *Integrated analysis of multimodal single-cell data*. *Cell*, 2021. **184**(13): p. 3573-3587 e29.

REVIEWERS' COMMENTS

Reviewer #1 (Remarks to the Author):

The changes have greatly improved the manuscript, and the authors' attention to recommended edits and suggestions is appreciated. In the context of the very thorough responses to reviewer comments, there are only stylistic suggestions, but overall this reviewer supports publication.

Suggestions:

1. It is recommended that the authors tone down language or soften claims associated with clinical stratification based on lineage scoring relating to Figure S2C because the score appears to give a continuous distribution with respect to FP-RMS and FN-RMS instead of clearly separating the subtypes into two groups. On page 9, end of the 1st paragraph, "Also, of note, the FP-" Indicates that samples are "stratified" but perhaps the phrasing could be modified to "were associated" with in the context of the core signature scores or "separates on a continuous spectrum". Additionally, it would be clarifying to soften any language that implies the muscle lineage scores clearly delineates FP- and FN-RMS, given the continuous distribution observed for the score.
2. It is very interesting that there is evidence of neuronal gene expression in fusion-positive RMS, but in some cases there is apparent overlap between populations. Specifically, in the rebuttal page 8,9 on Figure S4A, there is some overlap in gene expression profile between the neuronal and differentiated populations. Additionally in Figure 4E it appears that the neuronal population can map to muscle cell lineages. It would be helpful to provide some softening or clarifying language in the text to indicate that there is overlap in some cases.
3. In the abstract (Summary) the first sentence the term "ascribed" may be connotative of "described" and the authors should consider making the update if applicable to the intended meaning.
4. In Figure 1A, the upper right panel the term "unified" is slightly obstructed as rendered and could be realigned to clarify.

Reviewer #2 (Remarks to the Author):

The authors did a great job addressing this reviewers questions/concerns and have supplied the new information in the manuscript or discussion as requested. Really nice work.

Reviewer #3 (Remarks to the Author):

The reviewers adequately addressed my comments.

Reviewer #3 (Remarks on code availability):

A very nice addition to the work and a tool for future analytic teams.

RE: NCOMMS-23-51656-T

“Single cell transcriptomic profiling identifies tumor-acquired and therapy-resistant cell states in pediatric rhabdomyosarcoma”

Point-by-point response to reviewer comments

We thank the reviewers for their thorough and constructive comments. We have addressed the changes recommended by reviewer 1 within the main text of the manuscript. Please see below for our point-by-point response.

REVIEWER #1:

The changes have greatly improved the manuscript, and the authors' attention to recommended edits and suggestions is appreciated. In the context of the very thorough responses to reviewer comments, there are only stylistic suggestions, but overall this reviewer supports publication.

Authors' response: We appreciate the thorough comments, and have incorporated them into our final manuscript draft.

1. It is recommended that the authors town down language or soften claims associated with clinical stratification based on lineage scoring relating to Figure S2C because the score appears to give a continuous distribution with respect to FP-RMS and FN-RMS instead of clearly separating the subtypes into two groups. On page 9, end of the 1st paragraph, “Also, of note, the FP-” Indicates that samples are “stratified” but perhaps the phrasing could be modified to “were associated” with in the context of the core signature scores or “separates on a continuous spectrum”. Additionally, it would be clarifying to soften any language that implies the muscle lineage scores clearly delineates FP- and FN-RMS, given the continuous distribution observed for the score.

Authors' response: As the reviewer has suggested, we have edited the text to soften the language about our lineage scoring method (modified text is underlined):

“Also of note, the FP- ($n = 93$ genes) and FN-RMS ($n = 67$ genes) core signatures reported by Wei et al. [21] also separated these two subtypes along a continuous spectrum (Figure S2C), suggesting underlying gene program differences between these two subtypes of tumors.” (pg. 9)

2. It is very interesting that there is evidence of neuronal gene expression in fusion-positive RMS, but in some cases there is apparent overlap between populations. Specifically, in the rebuttal page 8,9 on Figure S4A, there is some overlap in gene expression profile between the neuronal and differentiated populations. Additionally in

Figure 4E it appears that the neuronal population can map to muscle cell lineages. It would be helpful to provide some softening or clarifying language in the text to indicate that there is overlap in some cases.

Authors' response: We have softened the language in our manuscript as the reviewer suggested (removed text in ~~strikethrough~~):

“...a ~~transcriptionally-distinct~~ tumor subpopulation in FP-RMS tumors that expresses neuronal marker genes including DCX, L1CAM, SYP, and CHGA (‘neuronal’ cluster; 1.4% and 4.8% of total cells/nuclei from PAX3::FOXO1 and PAX7::FOXO1 FP-RMS, respectively, **Figures 4A, S4A and S4B**).” (pg. 11)

In the Discussion, we have added a sentence to highlight the open question of the relationship between myogenic and neuronal gene expression programs:

“Moreover, future work is needed to clarify the relationship between the myogenic and neuronal cell states, and the mechanism by which FP-RMS tumors adopt the neuronal state.” (pg. 21)

3. In the abstract (Summary) the first sentence the term “ascribed” may be connotative of “described” and the authors should consider making the update if applicable to the intended meaning.

Authors' response: As the reviewer suggested, we have replaced the word “ascribed” with “described” in the abstract (modified text is underlined):

“...yet the extent to which cell state heterogeneity is shared with human development has not been described.” (pg. 2)

4. In Figure 1A, the upper right panel the term “unified” is slightly obstructed as rendered and could be realigned to clarify.

Authors' response: Thank you for catching the error. We have corrected Figure 1A to avoid truncation of the text.

REVIEWER #2:

The authors did a great job addressing this reviewers questions/concerns and have supplied the new information in the manuscript or discussion as requested. Really nice work.

Authors' response: Thank you for the kind comments.

REVIEWER #3:

The reviewers adequately addressed my comments.

A very nice addition to the work and a tool for future analytic teams.

Authors' response: Thank you for the kind comments.